# Seroepidemiology of enterovirus A71 infection in prospective cohort studies of children in southern China, 2013-2018

Juan Yang [1,14], Qiaohong Liao[1,2,14], Kaiwei Luo[3,14], Fengfeng Liu[2,14], Yonghong Zhou [1,14], Gang Zou[4,14], Wei Huang[3], Shuanbao Yu[2], Xianglin Wei[1], Jiaxin Zhou[1], Bingbing Dai[5], Qi Qiu[1], Ralf Altmeyer[4,6], Hongan Hu[5], Juliette Paireau [7,8], Li Luo[2], Lidong Gao[3], Birgit Nikolay[7], Shixiong Hu[3], Weijia Xing[2,9], Peng Wu[10], H. Rogier van Doorn[11,12], Peter W. Horby [13], Peter Simmonds[11], Gabriel M. Leung [10], Benjamin J. Cowling[10], Simon Cauchemez [7] & Hongjie Yu [1] ✉

Enterovirus A71 (EV-A71)–related hand, foot, and mouth disease (HFMD) imposes a substantial clinical burden in the Asia Pacific region. To inform policy on the introduction of the EV-A71 vaccine into the National Immunization Programme, we investigated the seroepidemiological characteristics of EV-A71 in two prospective cohorts of children in southern China conducted between 2013 and 2018. Our results show that maternal antibody titres declined rapidly in neonates, with over half becoming susceptible to EV-A71 at 1 month of age. Between 6 months and 2 years of age, over 80% of study participants were susceptible, while one third remained susceptible at 5 years old. The highest incidence of EV-A71 infections was observed in children aged 5-6 months. Our findings support EV-A71 vaccination before 6 months for birth cohorts in southern China, potentially with a one-time catch-up vaccination for children 6 months-5 years old. More regionally representative longitudinal seroepidemiological studies are needed to further validate these findings.

Hand, foot, and mouth disease (HFMD), a common childhood illness caused by enteroviruses, causes a substantial disease burden in the Asia Pacific region, particularly in mainland China[1,2]. The national HFMD surveillance system in China has shown that enterovirus A71 (EV-A71) is one of the most common serotype among reported laboratory-confirmed HFMD cases, especially in severe cases with neurological complications and/or cardiopulmonary involvement and fatal cases[2,3]. Additionally, a previous study described the substantial hospitalization burden of EV-A71–associated HFMD in southern China[4]. However, these studies were based on surveillance and health-care utilization data, which may have underestimated the infection rate of EV-A71 given that 29–71% of EV-A71 infections in young children are estimated to be very mild or asymptomatic[5,6]. Although these studies are important to quantify the clinical burden of EV-A71-

associated HFMD, they cannot provide a full understanding of the epidemiology and transmission of EV-A71.

Although no antiviral against EV-A71 is currently available, three inactivated monovalent EV-A71 vaccines have been licenced for children aged 6–71 months of age in China[7,8]. Since 2016, the vaccine has been used in the private sector in China but is not included in the National Immunization Programme, and the cost of vaccination is completely borne by individuals. The Chinese Center for Disease Control and Prevention (China CDC) recommends two doses of EV-A71 vaccination for children aged 6 months to 5 years who are susceptible to EV-A71 with a one-month interval, and encourages completing two doses before 12 months of age[9]. However, the uptake rate is much lower than that of vaccines introduced into the National Immunization Programme (greater than 90%)[10]. The uptake rate ranges from 1.3%

(Shantou in Guangdong province)[11] to 35.1% (Shanghai)[12] in developed cities. At the national level, the estimated vaccination coverage rate was between 2016 and 2020 at a maximum of 17.6% and 9.3% for children 6 months-2 years and 3–5 years old, respectively (Supplementary Table 1). Of all doses of EV-A71 vaccines, only 3% were used in children 3–5 years old[13].

Age-specific proportions of susceptible individuals, incidence rates of EV-A71 infections in children, and the dynamics of GMT after natural infection are required to inform policy-making about the introduction of EV-A71 vaccine into the National Immunization Programme. More specifically, these data are necessary to optimize the vaccination programme, identify target populations, determine whether a one-time catch-up vaccination is necessary for specific age groups, and determine whether naturally infected children require vaccination.

Recent studies of EV-A71 infections have mostly relied on cross-sectional serosurveys, which can identify prevalence but cannot identify recent versus historical infections[14–18]. To our knowledge, only a few prospective cohort studies have characterized the seroepidemiology of EV-A71 infections[6,19–22], and most have studied and followed neonate cohorts for only very short periods of time (≤one year). Our previous longitudinal cohort study with mother–neonate pairs assessed the antibody kinetics from birth to age 3 years, but could not address the longer-term pattern[23].

In this study, we therefore conducted a longitudinal cohort study in 2013 to gain insight into the epidemiology of EV-A71 in young children in southern China across multiple years and integrated it with the aforementioned neonate cohort[23]. We document the seroprevalence and occurrence of new infections of EV-A71 each season, and quantify the dynamics of neutralizing antibodies after natural infection from 0 to 12 years old.

## Results

### Study participants
A prospective, population-based cohort of children aged 1–9 years old (Children cohort) was established in Anhua County, China from 23 September 2013 to 25 November 2013. We approached 5996 children aged 1-9 years, and 4188 (70%) were enrolled in the study. Of these, 50.5% (2115/4188) were boys. Regular follow-up visits were conducted between August and November every year (hereafter annual visit) for all enrolled participants during 2014–2016. And an average of 25% of enrolled participants in each age group (hereafter subgroup) were randomly selected to additionally participate in three follow-up visits between February and March during 2014 and 2016 (hereafter semi-annual visit). At each visit, a venous blood sample (2 ml) was drawn from the participant. (See details in Methods, and Supplementary Methods Tables 2–4).

Figure 1 describes the enrolment, follow-up visits, and lab tests that were performed in the Children cohort. The proportion of participants attending each annual follow-up exceeded 80%. Neutralizing antibodies against EV-A71 were measured in 59% of participants ($n = 2475$). The baseline characteristics of the 2475 individuals in Children cohort are shown in Table 1. Supplementary Table 5 compares the baseline characteristics of participants with and without lab tests being performed. The two populations in Children cohort were similar in terms of sex, ethnicity, and annual family income, but older children were overrepresented in the group that was tested. Of the 2475 individuals in Children cohort, 50.3% (1246/2475) were boys. The baseline characteristics of boys were similar to that of girls, except birth weight (Supplementary Table 6).

In addition, a longitudinal, paired mother–neonate cohort (Neonates cohort) was established from 23 September 2013 to 14 October 2015. A total of 1066 newborns were enrolled. Of these, 54.0% (576/1066) were boys. The follow-up rate was over 50% at each visit, i.e., at ages 2, 4, 6, 12, 24, and 36 months. In addition to

cord blood, a venous blood sample was drawn from the participant at each visit. The flowchart of recruitment and follow-up visits, and the baseline characteristics of Neonates cohort have been described previously[23].

### Seroprevalence and annual infection of EV-A71
The time series of EV-A71 activity during 2013–2016 in Anhua County[23] and the follow-up periods are depicted in Supplementary Fig. 1. The time series of EV-A71 activity during 2013–2016 in Anhua County presented a clear peak in EV-A71 cases in March-June 2014 and April-July 2016. A smaller peak was detected in April–June 2015[23]. In Children cohort, EV-A71 seroprevalence did not vary significantly between consecutive semi-annual visits occurring before or after a peak of EV-A71 activity (e.g., baseline and follow-up visit 1), and few infections were detected during these time periods (Supplementary Figs. 2 and 3). We thus only present results for annual visits for seroprevalence and incidence of EV-A71 infections. Seropositivity was defined as a titre of 16 or greater in the main analysis. New infection with EV-A71 was conservatively defined as an individual whose titres moved from below to above the cutoff.

In Children cohort, as study participants aged during the course of the study, the seroprevalence in study participants gradually increased from 57% (95% CI: 55–59%) in 2013 to 73% (71–75%) in 2016 (Table 2). The incidence of EV-A71 infections in study participants was higher in 2015/16 (7.8%, 95% CI: 6.7–9.2%) than in 2013/14 (5.4%, 4.5–6.5%) and 2014/15 (5.4%, 4.5–6.5%) (Table 2). After correcting for the ageing of the study population, we found that seroprevalence in children aged 4-9 years old was roughly stable over time, ranging from 75% (70–80%) in 2013 to 70% (66–75%) in 2016, whereas the incidence of EV-A71 infections in children aged 4–9 years old in Anhua County remained higher in 2015/16 compared with previous seasons (6% vs. 3%). The seroprevalence and incidence standardized using the age structure of Anhua County are shown in Table 2. The threshold of seropositivity had little impact on the estimates of seroprevalence and incidence of EV-A71 infections (Supplementary Tables 7-8).

The proportion of susceptible individuals (i.e., individuals whose titres were less than 16) for EV-A71 demonstrated an inverted V-shape before 5 years old (Fig. 2a). As stated in a previously published study[23], neonates acquired protective concentrations of EV-A71 antibodies from mothers. As the maternal antibody titres declining rapidly, the susceptible individuals accumulated in the population. After the reduction in maternal antibody titres, the proportion of susceptible individuals quickly decreased due to natural infection. As demonstrated in the fitted curve presented in Fig. 2a, the proportion of neonates who were susceptible to EV-A71 increased from 26.5% (95% CI: 23.0–30.3%) at birth to 56.3% (52.6–59.9%) at 1 month of age, 75.9% (72.7–78.8%) at 2 months of age, and over 90% after 5 months of age. Approximately 85.7–93.0% of study participants were susceptible to EV-A71 between 6 months and 2 years old. A total of 63.8% (95% CI: 60.6–67.0%) and 34.4% (31.1–37.8%) of the study participants remained susceptible at the ages of 3 and 5 years old, respectively. Afterwards, the proportion of susceptible individuals declined slowly and tended to level off.

High incidence rates of EV-A71 infection were observed before 5 years of age, which showed an increasing trend before 6 months of age and then declined, but remained very high in children aged 3–5 years old (Fig. 2b). Particularly, the risk of new infection with EV-A71 among older children in the years with larger EV-A71 activity (such as the 6–7 year age group in 2015/2016[23]) was comparable to or even greater than that in younger children in the years with lower EV-A71 activity (such as the 0–2 age group in 2014/2015[23]) (Fig. 2b). Similar pattern was observed for incidence rates of EV-A71–related HFMD, which increased over age and peaked at 2 years old, and then declined but remained high at 3-5 years old (Fig. 2c).

No differences in the proportion of susceptible populations and new infections of EV-A71 (a cutoff titre of 16) were observed between boys and girls (Supplementary Fig. 4).

## Dynamics of neutralizing antibody titres of EV-A71

Starting with maternal immunity, a V-shape was observed for neutralizing antibody titres of EV-A71 before 5 years old in all study participants, and titres tended to level off afterwards (Fig. 3a). Geometric mean titre (GMT) of maternal EV-A71 antibody titres declined rapidly from 22.7 to below the protective titre of 16 in 16 days, and to the lowest at the age of 7 months (Fig. 3a). Then GMT gradually increased to over 16 by 30 months of age due to natural infection (Fig. 3a)[23]. Among all seropositive individuals, after natural infection other than vaccination, GMT declined from 737 (95% CI: 520–1044) to 80 (95% CI: 62–103) with age (Fig. 3b). Nonetheless, antibody titres remained greater than 64 among children aged 12 years old (Fig. 3b).

For all naturally infected individuals at baseline, the probability of returning to being susceptible increased with age but remained low at the age of 11 years old (7.2%, 95% CI: 5.2–9.1%). A higher probability occurred in those individuals whose initial antibody titres were <128 (5.9% [95% CI: 3.4–8.2%] at 5 years old with a peak of 22.5% [16.5–28.0%] at 11 years old) compared to those whose initial antibody titres were ≥128 (0.2% [95% CI: 0–0.5%] at 5 years old with a peak of 1.2% [0–2.3%] at 7 years old) (Fig. 3c). Sensitivity analyses showed that lower initial titres would lead to a large increase of the probability. For instance, in those individuals whose initial antibody titres were <64, the probability increased to 53% (95% CI: 38–65%) at the age of 11 years old (Supplementary Fig. 5).

Similar patterns of age-specific GMT were observed between boys and girls. No differences in the probability of returning to be susceptible to EV-A71 were observed between these two groups (Supplementary Fig. 6).

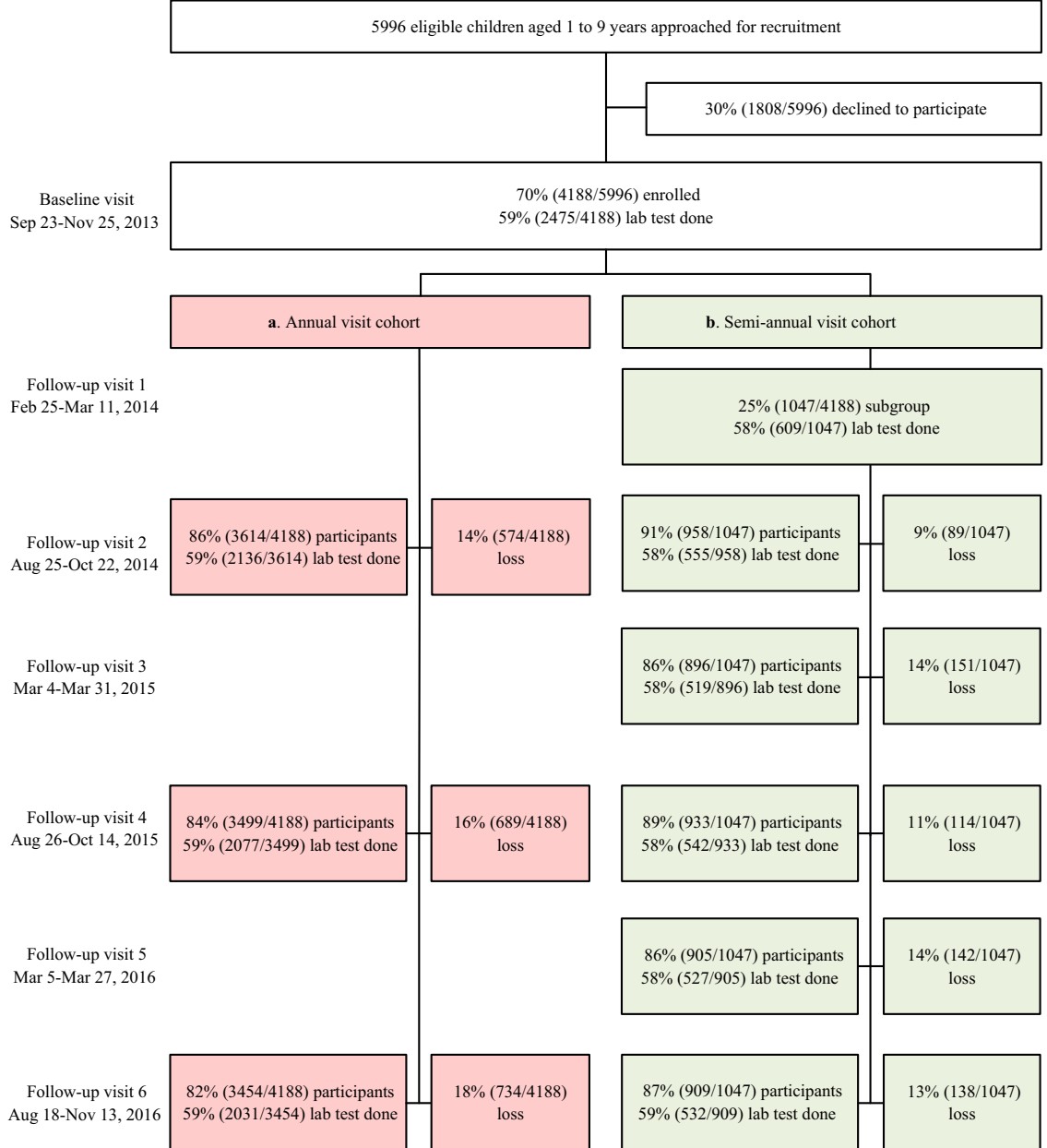

**Fig. 1 | Flowchart of recruitment for all participants aged 1–9 years and the follow-up visit rates (N = 4188). a** Regular follow-up visits between August and November every year (i.e., annual visit) for all enrolled participants during 2014–2016. **b** The semi-annual visit, i.e., an average of 25% of enrolled participants in each age group were randomly selected to additionally participate in three follow-up visits between February and March during 2014 and 2016.

**Table 1 | Characteristics of participants with lab test results (n = 2475) being performed at recruitment in Children cohort**

| Characteristics | No. participants | % |
|---|---|---|
| **Male, sex** | 1246 | 50 |
| **Age at baseline (year)** | | |
| 1 | 355 | 14 |
| 2 | 368 | 15 |
| 3 | 332 | 13 |
| 4 | 333 | 13 |
| 5 | 330 | 13 |
| 6 | 283 | 11 |
| 7 | 205 | 8 |
| 8 | 162 | 7 |
| 9 | 107 | 4 |
| **Ethnicity (Han)** | 2474 | 100 |
| **Township** | | |
| Tianzhuang (1.5 persons per 10⁴ m²) | 586 | 24 |
| Jiangnan (2.1 persons per 10⁴ m²) | 787 | 32 |
| Qingtang (2.7 persons per 10⁴ m²) | 1102 | 45 |
| **Birth mode*** | | |
| Full-term birth | 2344 | 95 |
| Preterm birth | 105 | 4 |
| Post-term birth | 25 | 1 |
| **Delivery mode*** | | |
| Natural childbirth | 1577 | 64 |
| Caesarean | 897 | 36 |
| **Birth weight (kilograms)** | 3.2 (0.5)# | / |
| **Underlying diseases*†** | 16 | 1 |
| **Number of household members** | | |
| 2 | 64 | 3 |
| 3 | 431 | 17 |
| 4 | 695 | 28 |
| 5 | 531 | 21 |
| ≥6 | 754 | 30 |
| Parents | 1325 | 54 |
| Others | 1150 | 46 |
| **Annual family income (RMB, Yuan)** | | |
| <20,000 | 611 | 25 |
| [20,000, 50,000) | 1382 | 56 |
| ≥50 000 | 482 | 19 |

"*" That total No. assessed were 2474 participants; "#" indicates mean (standard deviation); "†" indicates that there were 15 participants with congenital heart disease and 1 with oesophageal atresia.

## Discussion

This study quantifies the seroprevalence and incidence of EV-A71 infection over multiple seasons, and depicts patterns of susceptibility and neutralizing antibody concentrations among children aged 0–12 years old in southern China. This finding demonstrates that EV-A71 infection is common in children, and the proportion of susceptible individuals accumulated rapidly among those under 6 months of age as the maternal antibody titres declining, i.e., over a half of children were susceptible to EV-A71 at 1 month of age. Afterwards, increased natural infections lead to a decline of susceptible children, but over one-third of children remained susceptible at 5 years old. The highest incidence of EV-A71 infections was observed in children 5-6 months of age. The neutralizing antibody titres decline with age among naturally infected children. Our study also reveals that naturally infected individuals have a low

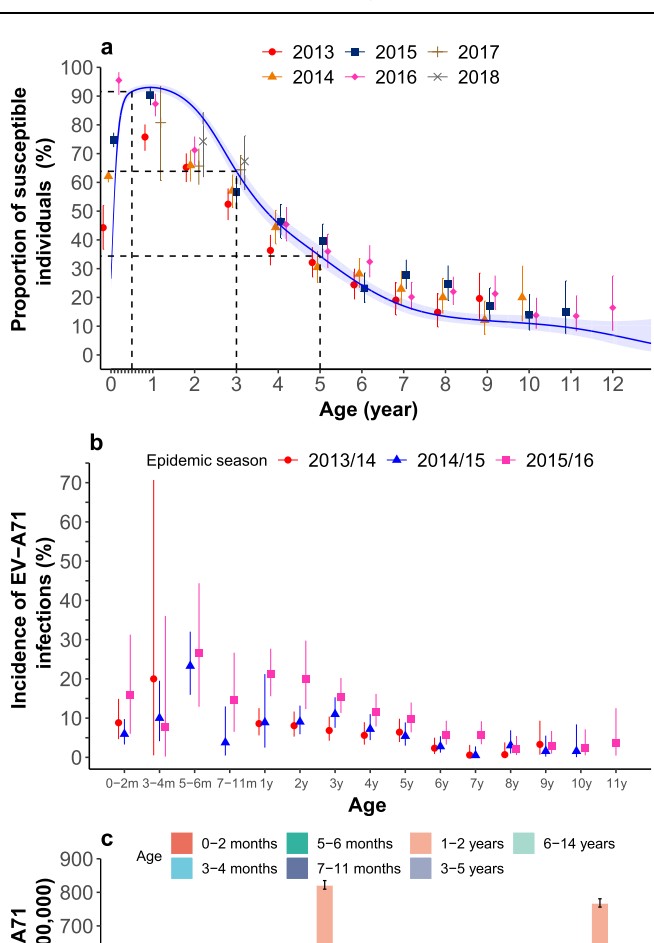

**Fig. 2 | The proportion of susceptible populations and new infections of EV-A71 (a cutoff titre of 16) and incidence of EV-A71–related hand, foot, and mouth diseases by age and season. a** Age-specific proportion of susceptible populations for all study participants in Children and Neonate cohort (points represent observed mean proportion; blue curve represents fitted mean proportion, whereas blue shadow represents corresponding 95% CI; the ticks between label 0 and 1 on x-axis represent 1–11 months of age). **b** Age-specific incidence of EV-A71 infections identified by serology for those study participants with paired sera before and after HFMD epidemics[23]. In the x-axis, "m" represents age in months, "y" represents age in years. Error bar represents corresponding 95% CI. *Age at the first blood draw for the paired sera. **c** Incidence of EV-A71-associated hand, foot, and mouth diseases in Hunan province by age group. Error bar represents corresponding 95% CI. Note: the sample size of each panel was listed in Supplementary Table 9.

probability of returning to being susceptible to EV-A71 before 6 years old, but lower initial titres would lead to a large increase of the probability.

The highest infection rates in children aged 0–2 years may be explained by the decay of maternal antibody[20,22,24] and immunologic naivety to EV-A71. The relatively high infection rates in children aged 3-5 years might be associated with immunologic naivety to EV-A71, as shown in our study, and an increased frequency of contact with other children due to admission to school. Altogether, the high infection

**Table 2 | Seroprevalence and incidence of EV-A71 infections by serology in study participants of the Children cohort and that standardized using the age structure of the children in Anhua County (a cutoff titre of 16)**

| | Age of study participants | Seroepidemiological outcomes of study participants | Seroepidemiological outcomes standardized using the age structure of the children in Anhua County |
|---|---|---|---|
| **Seroprevalence** | | | |
| Baseline visit, 2013 | Aged 1–9 years | 57% (55–59%) | 61% (58–65%) |
| Follow–up visit 2, 2014 | Aged 2–10 years | 62% (60–64%) | 66% (62–70%) |
| Follow–up visit 4, 2015 | Aged 3–11 years | 67% (65–69%) | 70% (66–74%) |
| Follow–up visit 6, 2016 | Aged 4–12 years | 73% (71–75%) | 75% (71–79%) |
| **Incidence of EV-A71 infections** | | | |
| 2013/14 | Aged 1–9 years at baseline visit in 2013 and 2–10 years at the follow-up visit 2 in 2014 | 5.4% (4.5–6.5%) | 4.8% (4.0–5.9%) |
| 2014/15 | Aged 2–10 years at the follow-up visit 2 in 2014 and 3–11 years at the follow-up visit 4 in 2015 | 5.4% (4.5–6.5%) | 4.8% (3.9–6.0%) |
| 2015/16 | Aged 3–11 years at the follow-up visit 4 in 2015 and 4–12 years at the follow-up visit 6 in 2016 | 7.8% (6.7–9.2%) | 6.8% (5.7–8.3%) |

rates in children <6 years suggest that this population is key to controlling EV-A71–related HFMD.

Current EV-A71 C4 genotype-based vaccines are approved for children aged 6–71 months in mainland China. China CDC recommends children aged 6 months being administered as early as possible, and encourages completing two doses before 12 months of age. Our study showed that as maternal EV-A71 antibody declined rapidly, over half of children were susceptible to EV-A71 at 1 month of age, while the proportion increased to three-quarters at 2 months of age and then peaked after 5 months of age. Correspondingly, incidence of EV-A71 infections and EV-A71–related HFMD showed an increasing trend before 6 months of age, and incidence of EV-A71 infections peaked at 5–6 months of age. Our previous study demonstrated that the time to loss of protective immunity of maternal antibody was less than two months[23]. Moreover, children younger than 6 months have higher risk of severe outcomes (e.g., case-fatality risk: about 0.17% in children younger than 6 months vs 0.11% in those aged 6–11 months) given infections relative to older children[2]. Altogether, administering the first doses at age 6 months might be too late to protect infants younger than that.

A new EV-A71 B4 genotype-based vaccine developed in Taiwan which can be administered to infants as young as 2 months of age appears to be safe, well-tolerated, and almost 100% effective against EV-A71-associated diseases[25]. We strongly recommend further investigating the optimal vaccination timing of approved EV-A71 vaccines in mainland China as early as 1–2 months of age if introduced into the National Immunization Programme[23], through further assessment of the dosing, safety, and effectiveness. In addition, if EV-A71 vaccines are introduced into the National Immunization Programme for children younger than 6 months of age on the basis of a birth cohort, a one-time catch-up vaccination for children 6 months-5 years of age is highly recommended, accounting for their high risk of EV-A71 infection and EV-A71-related HFMD due to susceptibility and increasing frequent contact with other children, as well as high case-severity risk and case-fatality risk[2].

Our findings reveal that naturally infected individuals have a low probability of returning to a susceptible status. From this perspective, the findings support China CDC's recommendation for the target population of EV-A71 vaccination. That is, susceptible individuals (namely, those without a protective neutralizing antibody titre against EV-A71–related diseases) aged 6 months-5 years[9]. Hereafter it is referred to a risk-based vaccination recommendation. However, in naturally infected individuals whose initial antibody titres were <128, the probability of returning to being susceptible increased to 22.5% at 11 years old. The probability would even increase to over 50% if initial antibody titres were <64. The antibody levels would be maintained via potential frequent re-exposures to EV-A71. Accordingly, the probability of returning to being susceptible would be underestimated. However, we were unable to evaluate the impact of potential re-exposures on susceptibility due to limited follow-up visits and long intervals between follow-up visits. In addition, the risk-based vaccination recommendation would have a negative impact on the implementation of vaccination programmes (i.e., facing the challenge of identifying susceptible individuals) and thus on vaccine coverage[26]. To increase vaccine uptake and ensure longer protection, we strongly recommend age-based vaccination instead of risk-based vaccination for EV-A71.

Currently, the China CDC does not recommend EV-A71 vaccines for children > 5 years old[9]. Our study reveals that older children still have a risk of infection with EV-A71, and their risk in large epidemic years is even comparable to that of younger children in years with low EV-A71 activity. Previous studies demonstrated that approximately 10% of EV-A71–related HFMD cases occurred in children >5 years old with a case-severity risk of 0.5–1.0%[2,3], and approximately 50 hospitalizations per 100,000 were attributed to EV-A71-related HFMD among children aged 5-9 years[4]. Increasing vaccine uptake among the eligible population (6–71 months) may provide indirect protection for older children. Whether to directly vaccinate older children or not merits further research. Systematic and continuous monitoring of the disease burden of EV-A71, vaccine safety and efficacy/effectiveness, and the health and economic impact of potential EV-A71 vaccination is also warranted.

Our study was strengthened by the cohort sequential longitudinal design with a large sample size and high follow-up rates. The older age groups in our study are important to determine the duration of immunity and to support future research on transmission modelling. However, we were not able to quantify the proportion of infections that cause clinically significant illness because this would have required frequent regular active surveillance of each participant for episodes of HFMD throughout follow-up.

In southern China, two HFMD peaks are observed per year (in spring and autumn), whereas only one peak is observed in summer in northern China[2]. EV-A71 has been circulating in the Asian-Pacific

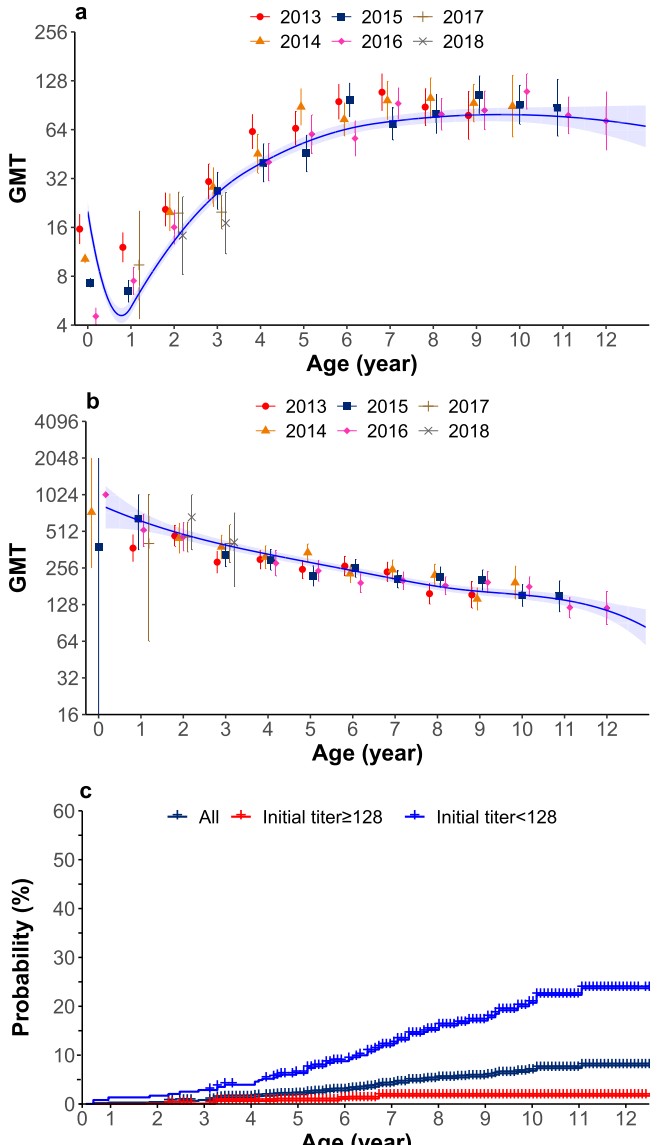

**Fig. 3 | EV-A71 geometric mean titres (GMT) with age and probability of returning to be susceptible to EV-A71. a** GMT for all study participants in Children and Neonates cohort. Dots and line represent observed and predicted GMT, respectively; error bars and shadow represent 95% CI. **b** GMT in the seropositive participants, excluding maternal antibody titres. Dots and line represent observed and predicted GMT, respectively; error bars and shadow represent 95% CI. **c** Kaplan–Meier plot of the probability of loss of immunity in participants who were infected. Note: the sample size of panel a and panel b was listed in Supplementary Table 9.

region since the 1990s. EV-A71 activity has remained at a low level in Europe and the USA for decades, but the number of EV-A71–related outbreaks has increased in the past several years[27–30]. Additionally, the predominant enterovirus serotype associated with HFMD varies between years[3,31]. EV-A71 was replaced by coxsackievirus A (CVA)16 or other enteroviruses serotypes that were the predominant serotypes for HFMD in several years (like 2013 and 2018) in China[3,31,32]. Given these different histories of EV-A71 exposure, it is not possible to generalize our estimates on seroprevalence and incidence of EV-A71 infections in southern China to northern China and other countries, as well as to other years outside the study period. Furthermore, a mathematical model study found that a potential high coverage of EV-A71 vaccination is likely to lead to transient and minor serotype replacement by CVA16[33]. Longer-term surveillance

of enteroviruses in larger geographic regions is therefore required to depict the seroepidemiological characteristics of EV-A71 and other enteroviruses, which could help selecting vaccine antigen, adjust and refine the vaccination strategy if necessary. Moreover, the development of multivalent vaccines could certainly contribute to prevention and control of HFMD[34].

Although neutralizing assays is currently used as the major test and widely accepted for neutralizing antibodies against EV-A71, some factors may have an impact on the reliability of the test and thus on the estimates of seroprevalence and incidence of EV-A71 infections. Firstly, cross-reactivity between EV-A71 and other enteroviruses may lead to false positive of EV-A71 infections. However, previous studies found that the sera from CVA16 infected patients were tested negative on an EV-A71 neutralization assay[35]. Following vaccination with inactivated EV-A71, seroconversion and protection is specific for EV-A71 infection, without similar neutralizing antibodies response and cross-protection observed for other enteroviruses like CVA16, CVA6 and CVA10[36–39].

Secondly, quality of laboratory testing procedure is critical for ensuring reliability of the test. As previous studies described[23,40], we took a series of measures for quality control of laboratory assays. For instance, each testing plate included two positive antibody control wells, two virus control wells, one serum toxicity control well for each test sample, and four cell control wells. Virus back titration was performed in each batch of test to determine the amount of attacking virus (within 32–320 $TCID_{50}/50\,\mu l$). EV-A71 neutralizing antibody standards (strongly positive, weakly positive and negative serum) from National Institutes for Food and Drug Control were used for quality control of serum antibody titre[41].

Thirdly, the choice of antibody titre threshold would influence the estimates of seroprevalence and incidence of EV-A71 infections. We used a cutoff of 16 in the main analysis, and added a cutoff of eight and 32 in the sensitivity analyses. We found that our results were robust to the choice of cutoffs.

Children younger than 6 years old represent the main susceptible population for EV-A71; thus, new infections largely occur among them, particularly in those younger than 6 months of age. This finding suggests in southern China, completing vaccination before 6 months of age would be beneficial, and we recommend age-based vaccination for birth cohorts. The findings that large proportions of children 6 months-5 years of age remained susceptible and their high incidence of EV-A71–related HFMD indicate a potentially one-time catch-up vaccination is of value in this age group. More regionally representative longitudinal studies, such as this study, that follow individuals with serial serology over multiple seasons for a longer period of time are needed to further validate these findings.

## Methods
A full description of the methods is provided in the Supplementary Methods. A summary is given below.

### Study setting and participants
A prospective, population-based cohort of children aged 1–9 years old (Children cohort) was established in three townships (Tianzhuang, Jiangnan, and Qingtang) in Anhua County in southern China. Participants were eligible for inclusion if they were 1–9 years old at enrolment, and resided in the study sites in the last ≥ 3 months. The 1-year age group was defined as those children aged 12–23 months; other age groups were defined in a similar manner.

Each study site generated a list of registered residents aged 1–9 years in the township who were eligible for enrolment. These individuals were then approached and invited by well-trained project personnel and/or village doctors in a randomized order. If selected children were unavailable or declined participation, the next eligible

child was approached. The process was iterated until a sufficient number of children were recruited in each age group.

Additionally, a longitudinal, paired mother–neonate cohort in the three townships was established from September 2013 to August 2018. The design of this cohort has been published previously[23]. Neonate participants of this cohort (Neonates cohort) were also included in this study. Of these, 54.0% (576/1066) were boys.

## Procedures
For Children cohort, a baseline blood draw was completed for all enrolled children at recruitment in September-November 2013. The dynamics of HFMD epidemics in the study area are characterized by two epidemic seasons per year with a larger seasonal peak in April–July and a smaller peak in late October-November[2,3,42]. Regular follow-up visits were conducted between August and November every year (hereafter annual visit) for all enroled participants during 2014–2016, producing paired sera samples to estimate the incidence rates of enterovirus infections for the annual epidemic season. To estimate the semi-annual incidence rates of enterovirus infections, an average of 25% of enrolled participants in each age group (hereafter subgroup) were randomly selected to additionally participate in three follow-up visits between February and March during 2014 and 2016 (hereafter semi-annual visit).

To summarize, in addition to the baseline visit, a total of six follow-up visits were conducted for the subgroup, separately between February-March 2014 (follow-up visit 1), August-October 2014 (visit 2), March 2015 (visit 3), August-October 2015 (visit 4), March 2016 (visit 5), and August-November 2016 (visit 6). Other participants only attended follow-up visits 2, 4 and 6. Those who participated in all annual visits and semi-annual visits are hereafter referred to as the full follow-up group. At each visit, a venous blood sample (2 ml) was drawn from the participant, and a questionnaire survey was completed by their caregivers.

For Neonates cohort, the detailed procedure has been published previously[23]. In addition to cord blood, infant blood samples were obtained at 2, 4, 6, 12, 24, and 36 months.

## Laboratory procedures
Completing the neutralizing assays on neutralizing antibodies against EV-A71 for all study participants is a resource-intensive task. For Children cohort, using multistage proportional stratified random sampling, we selected the specimens of 50% of enrolled participants aged 1–5 years for neutralizing assays. Considering the relatively smaller study population size in the 6- to 9-year-old age group, laboratory assays were conducted for all specimens in this age group. For Neonates cohort, the specimens of all participants were tested for neutralizing antibodies.

The EV-A71 strain (FY573, GenBank accession number: HM064456.1) used in this study was isolated from a child with HFMD from Fuyang city of Anhui province in 2008. Serum samples were inactivated at 56 °C for 30 min and then serially diluted 4-fold from 1:8 to 1:2048 with duplicate wells of each dilution. The detailed laboratory assay was reported previously[23]. Neutralizing titres were defined as the reciprocal of the highest dilution capable of inhibiting 50% of the CPE and calculated by the Karber method[43]. For the purposes of statistical analysis, neutralization titres <8 or >2048 were assigned a value of 4 or 4096, respectively. More details are provided in a previous publication[23].

## Enhanced surveillance of HFMD
National enhanced HFMD surveillance has been conducted across mainland China since May 2008[2]. The specimens were collected from all severe HFMD cases and the first 5 reported mild cases in each

county every month and tested for enteroviruses at local Centres for Disease Control and Prevention using PCR. Test results are characterized as (1) negative for enterovirus or positive for (2) EV-A71, (3) CVA16, or (4) other enteroviruses. Anhua County is part of Hunan Province, southern China. We obtained the surveillance data for Hunan province from the national enhanced HFMD surveillance for 2013–2016.

## Outcomes
By integrating Children cohort and Neonates cohort, we estimated the seropositivity of EV-A71 antibody and new infections with EV-A71. The protective titre of EV-A71 antibody has not been well characterized, but a phase 3 clinical trial of EV-A71 vaccines demonstrated that a titre of 16 could be considered as a possible serologic marker for protection against EV-A71-related HFMD[8]. Moreover, a previous study showed that the choice of antibody titre threshold (i.e., 8, 16 or 32) had minimal effect on the pattern for seropositivity[23]. Thus, in the main analysis, seropositivity was defined as a titre of 16 or greater. New infection with EV-A71 was conservatively defined as an individual whose titres moved from below to above the cutoff. Additionally, sensitivity analyses were done with a cutoff of eight (minimum detectable antibody level in neutralization assays) and 32.

We further described the age patterns for the proportion of susceptible individuals (i.e., individuals whose titres were less than 16), new infections and geometric mean titre (GMT) from 0–12 years old. Moreover, we estimated the probability of returning to being susceptible to EV-A71 after natural infection between 0–12 years old.

Using data from enhanced HFMD surveillance in Hunan province, we calculated the incidence rate of EV-A71–associated HFMD.

## Statistical analysis
All analyses were performed in R version 3.5.0 (R Foundation for Statistical Computing, Vienna, Austria, https://www.r-project.org/) and SAS 9.4 (SAS Institute Inc., Cary, NC, USA).

For Children cohort, we calculated that a sample size of 700 participants per age group for the 1-year and 2-year age groups and of 650 per age group for the other age groups would allow a 10% annual risk of EV-A71 infection to be estimated with a statistical significance level of 5%, a 2.5% marginal error, and a dropout rate of 21% for the 1-year and 2-year age groups and 15% for the other age groups. Similarly, for Neonates cohort, a sample size of 900 infants would allow a 10% annual risk of EV-A71 infection[23]. No study participants in Children cohort were administered EV-A71 vaccines during the study period. Seven study participants in Neonates cohort were administered EV-A71 vaccines after 6 months of age during the study period; thus, their antibody titres after vaccination were excluded from this analysis[23].

Seroprevalence was calculated for each visit and the probability of EV-A71 infection for each season. The calculation of incidence was restricted to those for whom paired serological samples (before and after HFMD epidemics[23]) were available. We used a binomial distribution to derive the 95% confidence intervals (using R package binom). The seroprevalence and incidence were then standardized to the age structure (1–9 years population) of Anhua County according to the 2013 National Bureau of Statistics Dataset.

We calculated GMT by age group for all participants in Children and Neonates cohort, and used t distribution to drive 95% confidence interval. To explore the dynamics of neutralizing antibody titres after natural infection, we further excluded the maternal antibody titres in Neonates cohort, and then calculated GMT for those with positive titres by age group. We applied generalized linear mixed models (PROC glimmix in SAS) using B-splines to fit the dynamics of the proportion of susceptible individuals and GMT. The model selection including B-spline's degree and knots, and model parameters were

based on Akaike Information Criterion (details shown in the Supplementary Notes). Moreover, using survival analysis (survfit function in R package survival), we estimated the probability of returning to being susceptible to EV-A71 after natural infection. This analysis was further stratified by initial antibody titres in order to explore the difference of immunity duration. We used an initial antibody titre of 128 in the main analysis, and conducted sensitivity analyses using 256, 64 or 32 instead.

## Ethics

Institutional review board approval was obtained from the Western Pacific Region Office of the World Health Organization (2013.10.CHN.2. ESR), China CDC (201224) and Fudan University (2019-05-0756), and written informed consent was obtained from all caregivers of participants.

## Reporting summary

Further information on research design is available in the Nature Portfolio Reporting Summary linked to this article.

## Data availability

The data that supporting the findings of this study are available on GitHub at https://github.com/JYoung2022FD/Sero-epi-data.git and the data provided are anonymised due to privacy/ethical restrictions. All other data can be obtained from the corresponding author upon request.

## Code availability

The code used to generate these analyses are available on GitHub at https://github.com/JYoung2022FD/Sero-epi-data.git.

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

## Acknowledgements

H.Y. acknowledges financial support from the Key Program of the National Natural Science Foundation of China (No. 82130093) and the Li Ka Shing Oxford Global Health Programme (No. LG33). G.Z. acknowledges financial support from the National Natural Science Foundation of China (grant number 31670159). L.G. acknowledges financial support from the Chinese Preventive Medicine Association (No. 20101801); the Hunan Provincial Natural Science Foundation (2019JJ80115); and the Scientific Research Project of Hunan Provincial Health Commission (B2019039). J.P., B.N. and S.C. acknowledge financial support from the AXA Research Fund, the Investissement d'Avenir program, the Laboratoire d'Excellence Integrative Biology of Emerging Infectious Diseases programme (Grant ANR-10-LABX-62-IBEID), the Models of Infectious Disease Agent Study of the National Institute of General Medical Sciences, and the INCEPTION project (PIA/ANR-16-CONV-0005). W.X. acknowledges financial support from the Young Taishan Scholars (No. tsqn20161046). P.S. and P.H. acknowledge financial support for collaborative work with H.Y. from the Wellcome Trust (ISSF204826/Z/16/Z). The funders had no role in the study design, data collection and analysis, decision to publish, or preparation of the manuscript.

We acknowledge that this work was a continuous research project conducted by H.Y., who was a former employee of the Chinese Center for Disease Control and Prevention and moved to Fudan University since May 2017.

We thank staff members of the Anhua County–, Yiyang Prefecture–, and Hunan Provincial–level departments of health for providing assistance with administration and data collection; staff members at the Anhua County–level (Liang Zhao, Zhiyong Chen, Min He, Zhaohui Jiang, and Ping Liu), Yiyang Prefecture–level (Lizhi Xie, and Qing Li), and Hunan Provincial–level Centres for Disease Control and Prevention (CDC) (Zhihong Deng, Fuqiang Liu, Zhifei Zhan, Liang Cai, Shanlu Zhao, Ge Zeng, Hao Yang, Yu Chen, Qianlai Sun, and Hengjiao Zhang), study hospitals (Anhua People's Hospital, Anhua Hospital of Traditional Chinese Medicine, Anhua Second People's Hospital, Tianzhuang Township Hospital, Jiangnan Township Hospital, and Qingtang Township Hospital) (Le Yang, Hong Liang, and Yi Zhang), China CDC (Yu Li, Junling Sun, and Chen Yuan), Hubei CDC (Qi Chen), and Mentougou District CDC (Xiangju Zhao) for providing assistance with field investigation, administration, and data collection. We thank Ke Lan, Qianqian Gao, Shilin Yuan, Ying Cui, Yang Li, Rui Cui, Xiaowei Mo, Guiming Li, Ying Wang and Xuemin Fu from the Institut Pasteur of Shanghai, Chinese Academy of Sciences, who helped with data collection and laboratory assays. We thank Yong Zhang, Shuangli Zhu from China CDC, Quanyi Wang, Jie Li from Beijing CDC, Teng Zheng, Jiayu Wang from Shanghai CDC; Tran Thuy Ngan and Le Van Tan from Oxford University Clinical Research Unit, Hospital for Tropical Diseases, Ho Chi Minh City; Qunying Mao and Zhenglun Liang from National Institutes for Food and Drug Control; Sirenda Vong from the World Health Organization Regional Office for South-East Asia; Jeremy J. Farrar from the Wellcome Trust, London; Joseph T. Wu from WHO Collaborating Centre for Infectious Disease Epidemiology and Control, School of Public Health, and Li Ka Shing Faculty of Medicine, The University of Hong Kong for providing technical support for this work. We also thank Jingbo Liang, Lingshuang Ren, Xinhua Chen, Jinxin Guo, Lili Wang, Shaolong Ren, Yangni He, Kai Wang, and Han Yan from the School of Public Health, Fudan University, Key Laboratory of Public Health Safety, Ministry of Education, Shanghai for helping with the laboratory assays, data collection and cleaning.

## Author contributions

H.Y. conceived, designed and supervised the study. J.Y., Q.L., K.L., F.L., S.Y., B.D., H.H., L.L., L.G., S.H., W.X. and P.W. coordinated and participated in data collection. Y.Z., G.Z., W.H., S.Y., and Q.Q. performed the lab test with methodological input from P.S. J.Y., Q.L., X.W., J.Z, J.P., and B.N. analyzed the data. J.Y. and X.W. prepared the figures. J.Y. prepared the first draft of the manuscript. R.A., J.P., B.N., R.D., P.H., P.S., G.M.L., B.J.C., S.C., and H.Y. commented on the data and its interpretation and revised the content critically. All authors contributed to the review and revision and approved the final manuscript as submitted and agree to be accountable for all aspects of the work.

## Competing interests

H.Y. has received investigator-initiated research funding from Sanofi Pasteur, GlaxoSmithKline, Yichang HEC Changjiang Pharmaceutical Company, Shanghai Roche Pharmaceutical Company and SINOVAC Biotech Ltd. None of the research funding is related to this study. All other authors report no competing interests.

## Additional information

[1]School of Public Health, Fudan University, Key Laboratory of Public Health Safety, Ministry of Education, Shanghai, China. [2]Key Laboratory of Surveillance and Early-warning on Infectious Disease, Division of Infectious Disease, Chinese Center for Disease Control and Prevention, Beijing, China. [3]Hunan Provincial Center for Disease Control and Prevention (Workstation for Emerging Infectious Disease Control and Prevention, Chinese Academy of Medical Sciences), Changsha, China. [4]Institut Pasteur of Shanghai, Chinese Academy of Sciences, Shanghai, China. [5]Anhua County Center for Disease Control and Prevention, Yiyang, China. [6]Medusa Therapeutics Limited, Hong Kong Special Administrative Region, Hong Kong, China. [7]Mathematical Modelling of Infectious Diseases Unit, Institut Pasteur, Université Paris Cité, UMR2000, CNRS, 75015 Paris, France. [8] Infectious Diseases Department, Santé publique France, Saint-Maurice, France. [9]School of Public Health, Shandong First Medical University & Shandong Academy of Medical Sciences, Tai'an, China. [10]WHO Collaborating Centre for Infectious Disease Epidemiology and Control, School of Public Health, Li Ka Shing Faculty of Medicine, The University of Hong Kong, Hong Kong Special Administrative Region, Hong Kong, China. [11]Oxford University Clinical Research Unit, Hospital for Tropical Diseases, Ho Chi Minh City, Vietnam. [12]Nuffield Department of Medicine, University of Oxford, Oxford, UK. [13]Centre for Tropical Medicine and Global Health, Nuffield Department of Medicine, University of Oxford, Oxford, UK. [14]These authors contributed equally: Juan Yang, Qiaohong Liao, Kaiwei Luo, Fengfeng Liu, Yonghong Zhou, Gang Zou. ✉e-mail: yhj@fudan.edu.cn

