## [Peer Review File · Nature Communications]

Seroepidemiology of enterovirus A71 infection in prospective cohort studies of children in Southern China, 2013-2018REVIEWER COMMENTS

Reviewer #1 (Remarks to the Author):

SUMMARY: Yang et al. report the seroprevalence and annual infection incidence of EV-A71 in two population-based cohorts of 4188 children cohort and 1066 neonates respectively, in order to inform policy-making for introduction of the EV-A71 vaccine into the National Immunization Programme of China. They found that 80% of children 6mo-2yrs were susceptible and that 69% and 26% remained susceptible at 3 and 5 years, leading them to recommend routine vaccination before 1 year of age, down to the approved 6 months of age. They also recommend universal single dose vaccination for children 2-5yo, instead of risk-based vaccination, for practical purposes to increase uptake and ensure longer protection. The longitudinal seroprevalence study conducted was large and labor-intensive and provides useful information to guide public policy. The data is well-analyzed and write-up is comprehensive with a balanced discussion.

MAJOR COMMENTS:

-While the authors focus on the group in which the Chinese EV-A71 vaccines are currently approved (6-71 months), the authors should further discuss the seroprevalence and infection data in the first year, including the first 6 months, of life to determine the ideal age for incorporation of the vaccine into the National Immunization Program. A recent publication of a Phase 3 trial of EV71vac (Nguyen et al. Lancet ID 2022), another EV-A71 vaccine developed in Taiwan, showed that this EV-A71 vaccine was safe and effective down to 2 months of age. They noted 27% of HFMD cases occurred in the 2-5 month age group that would have been unprotected if vaccine initiated at 6 months of age, and that this age group has had higher case severity and case fatality relative to older children. It would be helpful if the authors of this manuscript discussed the seroprevalence data in more detail from the first year of life, initially reported in the Wei et al. paper cited focusing on the Neonate cohort, particularly Fig 3D, which shows seroprevalence less than 50% at 2 months and 15% at age 4 months, and connect this with infection prevalence/seroconversion rate in these groups from the current study. While the current paper is very informative for incorporation of EV-A71 vaccinations into a national vaccine program, more detail should be provided as to ideal age of initial vaccination.

MINOR COMMENTS:

-Figure 1 could be more clear as to what is being displayed in the L-sided branch (n=3614) compared to the R-sided branch (n=1047) so the figure can stand alone.

-Table 1: since the denominator of total No. assessed is the same at n=2475 throughout, can list in header and remove column reporting for each characteristic.

Reviewer's report on: Seroepidemiology of children with enterovirus A71 infections in southern China: longitudinal, population-based cohort studies, 2013–2018

In this longitudinal cohort study, the authors aim to estimate the sero-prevalence and sero-incidence of enterovirus A71 (EV-A71) infection during 2003–2018 in rural areas of Southern China. EV-A71 is a frequent cause of severe hand, foot, and mouth (HFMD); a monovalent, inactivated vaccine against EV-A71 has been developed and is currently licensed in China. Its coverage, however, is currently limited, and this study is therefore intended to inform policy regarding its introduction into the national immunization program in China. The analysis is based on a large ($n = 2,475$) collection of blood samples in children aged 1–9 years, from which neutralizing antibodies were titrated to estimate the prevalence and the incidence of seropositivity. The results indicate a large burden of EV-A71 infection, especially in children <2 yr (with annual attack rates as high as 20% during some seasons).

Overall, this is a well-conducted and interesting study that provides useful estimates of the burden of EV-A71 infection. However, I think the study could be improved in a number of ways. In particular, the methods are currently insufficiently described to ensure reproducibility. More generally, I wonder if the estimates presented here are, in fact, sufficient to motivate universal vaccination, as recommended by the authors in their conclusion. More specific comments are listed below.

Major comments

1. The description of the methods is insufficient. For example, the authors state that “[...] applied generalized linear mixed models using B-splines to fit the dynamics of the proportion of susceptible individuals and GMT.” (lines 471–472). However, this is far too vague and more information is needed about the type of models used, the allowed wiggleness of the splines, the R packages used for fitting, etc. Similar comments apply to the survival analysis, whose results are presented in Figure 3C. Please also include a reproducibility statement, indicating (if possible) where to access the data and the computer codes to reproduce the results.
2. Overall, the study design seems sound, but I have a few questions:
 - (a) Were the estimates of sero-prevalence and sero-incidence corrected for imperfect sensitivity and specificity of the test?
 - (b) Did any child in the two cohorts receive the EV-A71 vaccine? If so, could that bias the estimates of sero-prevalence?
 - (c) In figure 3C, please justify the choice of the arbitrary cut-off titer of 128.
 - (d) To calculate sero-incidence, the authors write that: “New infection with EV-A71 was conservatively defined as an individual whose titers moved from below to above the infection cut-off.” (lines 174–176). Please also report this information in the methods, and explain how the confidence intervals for infection rates (Figure 2B) were derived. It would also be useful to assess how sensitive the estimates of infection rates were to the assumed cut-off.
3. From figure 3, the authors conclude that antibody decay and return to susceptibility are slow (expect for children whose initial antibody titers were low). However, a general limitation of sero-prevalence studies (even with detailed longitudinal follow-up as this one) is that not all events of re-exposure are observed. Hence, it could be that the “natural” decay rate of antibodies is quite high, but that apparently high levels of antibodies (as observed in figure 3B) are maintained via frequent re-exposures to EV-A71. This limitation may be especially important in the context of vaccination, as vaccines can reduce overall circulation in the population, and therefore also the frequency of re-exposure.
4. From a more general, public health perspective, I wonder if the study results are sufficient to support the authors' claim, as stated in the conclusions: “Our findings support completing vaccination before 1 year of age, and we recommend age-based universal vaccination for birth cohorts and a one-time catch-up vaccination for children 2-5 years of age.” (lines 331–334). First, there is an obvious extrapolation problem (acknowledged by the authors), as the study was only conducted in rural areas of Southern China and the epidemiology of EV-A71 may differ in other regions. Second, ultimately the decision to introduce a new vaccine should rely on other indicators, such as hospitalization or mortality rates. Figure 2C presents incidence rates of EV-A71-related severe HFMD, but the definition of “severe” in this case remained unclear, after re-reading the corresponding description (lines

416–424)—please clarify. Finally, one should be aware of the indirect effects of targeted vaccination, especially in the context of disease caused by polymorphic pathogens like EV. For example, one could imagine that the different EV types compete with each other, such that targeted vaccination against EV-71 could cause a release of other types. Not being in the field of EV epidemiology, I can't assess if this issue is important (or, indeed, relevant at all), but I think it should be, at least, discussed.

Minor comments

- The abstract would need to incorporate further information about the study design, period, and location. Please also report uncertainty intervals for the sero-prevalence estimates.
- p 4, lines 55–57: it seems these two sentences should not be separated by a full stop.
- p 13, l 247: I don't think the word “naivety” actually exists in English.

Reviewer #3 (Remarks to the Author):

In the abstract it would be helpful to state the data used. It isn't clear whether the incidence estimates are coming from the seroprevalence or different data. Also the study design is a strength of the paper, so good to highlight here!

Line 59-60: I think stating for what proportion of the population this is the case would be helpful.

Line 59: a word missing or incorrect tense I think

With the recommendation of vaccination at one year old, I think it would be good to highlight in the abstract the seropositivity at age 1.

Line 120-121, I think the phrasing about "rural areas" is maybe a little general without further discussion of how similar you think this transmission would be across different rural areas. Suggest either rephrasing here or adding more information on the (lack of) variation.

Line 217: Is this V shape hypothesized to start with maternal immunity, if so, please mention here. I understand this has been looked at by a previous paper by the authors, but I think also worth discussing here. Particularly with the conclusions around timing of vaccination- it will be important to vaccinate after the decline in maternal immunity too I would think.

Line 219, be useful to say natural infection, instead of what? Or how natural infection is defined.

Figure 2: Panel B: This is the estimated incidence from the serology? Please clarify.

General comments

1. Please comment on any/the lack of possible cross reactivity
2. Be helpful to have more comparison to the age distribution of cases and any different inferences that would be made on incidence of infection from the age distribution of cases
3. Figure 2 (Panel A): The older ages susceptible proportions don't seem to be captured so well by the model- what is happening here?

Reviewer #1 (Remarks to the Author):

1.1 Summary: Yang et al. report the seroprevalence and annual infection incidence of EV-A71 in two population-based cohorts of 4188 children cohort and 1066 neonates respectively, in order to inform policy-making for introduction of the EV-A71 vaccine into the National Immunization Programme of China. They found that 80% of children 6mo-2yrs were susceptible and that 69% and 26% remained susceptible at 3 and 5 years, leading them to recommend routine vaccination before 1 year of age, down to the approved 6 months of age. They also recommend universal single dose vaccination for children 2-5yo, instead of risk-based vaccination, for practical purposes to increase uptake and ensure longer protection. The longitudinal seroprevalence study conducted was large and labor-intensive and provides useful information to guide public policy. The data is well-analyzed and write-up is comprehensive with a balanced discussion.

Response: We would like to thank the reviewer for their constructive and positive comments.

1.2 Major comments: While the authors focus on the group in which the Chinese EV-A71 vaccines are currently approved (6-71 months), the authors should further discuss the seroprevalence and infection data in the first year, including the first 6 months, of life to determine the ideal age for incorporation of the vaccine into the National Immunization Program. A recent publication of a Phase 3 trial of EV71vac (Nguyen et al. Lancet ID 2022), another EV-A71 vaccine developed in Taiwan, showed that this EV-A71 vaccine was safe and effective down to 2 months of age. They noted 27% of HFMD cases occurred in the 2-5 months age group that would have been unprotected if vaccine initiated at 6 months of age, and that this age group has had higher case severity and case fatality relative to older children. It would be helpful if the authors of this manuscript discussed the seroprevalence data in more detail from the first year of life, initially reported in the Wei et al. paper cited focusing on the Neonate cohort, particularly Fig 3D, which shows seroprevalence less than 50% at 2 months and 15% at age 4 months, and connect this with infection prevalence/seroconversion rate in these groups from the current study. While the current paper is very informative for incorporation of EV-A71 vaccinations into a national vaccine program, more detail should be provided as to ideal age of initial vaccination.

Response: Thank you for pointing this out. The reviewer is correct that seroprevalence and infections in the first year, particularly before 6 months of age, are critical to determine the ideal age for incorporation of EV-A71 vaccine into the National

Immunization Program. As suggested, we have added more detailed analyses for the first year of life, by breaking down into 0-2, 3-4, 5-6, and 7-11 months of age groups.

In the Results section:

Page 12, lines 209-214: As demonstrated in the fitted curve presented in Fig. 2a, the proportion of neonates who were susceptible to EV-A71 increased from 26.5% (95% CI: 23.0%-30.3%) at birth to 56.3% (95% CI: 52.6%-59.9%) at 1 month of age, 75.9% (95% CI: 72.7%-78.8%) at 2 months of age, and over 90% after 5 months of age.

Page 12-13, lines 221-230: High incidence rates of EV-A71 infection were observed before 5 years of age, which showed an increasing trend before 6 months of age and then declined, but remained very high in children aged 3-5 years old (Fig. 2b)... Similar pattern was observed for incidence rates of EV-A71-related HFMD, which increased over age and peaked at 2 years old, and then declined but remained high at 3-5 years old (Fig. 2c).

Page 13, lines 239-243: Geometric mean titre (GMT) of maternal EV-A71 antibody titres declined rapidly from 22.7 to below the protective titre of 16 in 16 days, and to the lowest at the age of 7 months (Fig. 3a). Then GMT gradually increased to over 16 by 30 months of age due to natural infection (Fig. 3a).²³

Moreover, we have modified the Discussion section to discuss the ideal age of initial vaccination and catch-up vaccination if introducing the EV-A71 vaccine into the National Immunization Program.

Page 16-17, lines 289-318: Current EV-A71 C4 genotype-based vaccines are approved for children aged 6-71 months in mainland China. China CDC recommends children aged 6 months being administered as early as possible, and encourages completing two doses before 12 months of age. Our study showed that as maternal EV-A71 antibody declined rapidly, over half of children were susceptible to EV-A71 at 1 month of age, while the proportion increased to three quarters at 2 months of age and then peaked after 5 months of age. Correspondingly, incidence of EV-A71 infections and EV-A71-related HFMD showed an increasing trend before 6 months of age, and incidence of EV-A71 infections peaked at 5-6 months of age. Our previous study demonstrated that the time to loss of protective immunity of maternal antibody was less than two months²³. Moreover, children younger than 6 months have higher risk of severe outcomes (e.g., case-fatality risk: about 0·17% in children younger than 6 months vs 0·11% in those aged 6–11 months) given infections relative to older children². Altogether, administering the first doses at age 6 months might be too late to protect infants younger than that.

A new EV-A71 B4 genotype-based vaccine developed in Taiwan which can be administered to infants as young as 2 months of age appears to be safe, well-tolerated, and almost 100% effective against EV-A71 associated diseases²⁵. We strongly recommend further investigating the optimal vaccination timing of approved EV-A71 vaccines in mainland China as early as 1-2 months of age if introduced into the National Immunisation Programme, through further assessment of the dosing, safety, and effectiveness²³. In addition, if EV-A71 vaccines are introduced into the National Immunization Programme for children younger than 6 months of age on the basis of a birth cohort, a one-time catch-up vaccination for children 6 months-5 years of age is highly recommended, accounting for their high risk of EV-A71 infection and EV-A71-related HFMD due to susceptibility and increasing frequent contact with other children, as well as high case-severity risk and case-fatality risk².

Minor comments:

1.3 Figure 1 could be more clear as to what is being displayed in the L-sided branch (n=3614) compared to the R-sided branch (n=1047) so the figure can stand alone.

Response: Thank you pointing this out. We apologize for the lack of clarity. As suggested, we now split figure 1 into two panels: 1) panel a depicts regular follow-up visits between August and November every year (i.e., annual visit) for all enrolled participants during 2014-2016; 2) panel b depicts the semi-annual visit, i.e., an average of 25% of enrolled participants in each age group were randomly selected to additionally participate in three follow-up visits between February and March during 2014 and 2016. We have now clarified this point in the figure legend (Page 40, lines 869-874) in the revised text.

1.4 Table 1: since the denominator of total No. assessed is the same at n=2475 throughout, can list in header and remove column reporting for each characteristic.

Response: Thanks for pointing this out. We modified it in the revised text as suggested.

Reviewer #2 (Remarks to the Author):

2.1 Reviewer's report on: Seroepidemiology of children with enterovirus A71 infections in southern China: longitudinal, population-based cohort studies, 2013–2018

In this longitudinal cohort study, the authors aim to estimate the sero-prevalence and

sero-incidence of enterovirus A71 (EV-A71) infection during 2003–2018 in rural areas of Southern China. EV-A71 is a frequent cause of severe hand, foot, and mouth (HFMD); a monovalent, inactivated vaccine against EV-A71 has been developed and is currently licensed in China. Its coverage, however, is currently limited, and this study is therefore intended to inform policy regarding its introduction into the national immunization program in China. The analysis is based on a large ($n = 2,475$) collection of blood samples in children aged 1–9 years, from which neutralizing antibodies were titrated to estimate the prevalence and the incidence of seropositivity. The results indicate a large burden of EV-A71 infection, especially in children < 2 yr (with annual attack rates as high as 20% during some seasons). Overall, this is a well-conducted and interesting study that provides useful estimates of the burden of EV-A71 infection. However, I think the study could be improved in a number of ways. In particular, the methods are currently insufficiently described to ensure reproducibility. More generally, I wonder if the estimates presented here are, in fact, sufficient to motivate universal vaccination, as recommended by the authors in their conclusion. More specific comments are listed below.

Response: We would like to thank the reviewer for the many constructive comments they provided. In the revised manuscript, we have provided more details in Methods to ensure reproducibility, and we have modified the Results and Discussion to better motivate universal vaccination. Details are presented as follows.

Major comments

2.2 The description of the methods is insufficient. For example, the authors state that “[. . .] applied generalized linear mixed models using B-splines to fit the dynamics of the proportion of susceptible individuals and GMT.” (lines 471–472). However, this is far too vague and more information is needed about the type of models used, the allowed wiggleness of the splines, the R packages used for fitting, etc. Similar comments apply to the survival analysis, whose results are presented in Figure 3C. Please also include a reproducibility statement, indicating (if possible) where to access the data and the computer codes to reproduce the results.

Response: Thank you for pointing that out. The text has been updated as follows:

Page 28, Lines 552-554: All analyses were performed in R version 3.5.0 (R Foundation for Statistical Computing, Vienna, Austria, <https://www.r-project.org/>) and SAS 9.4 (SAS Institute Inc., Cary, NC, USA).

Page 29, Lines 580-587: We applied generalized linear mixed models (PROC glimmix in SAS) using B-splines to fit the dynamics of the proportion of susceptible

individuals and GMT. The model selection including B-spline's degree and knots, and model parameters were based on Akaike Information Criterion (details shown in the Supplementary Notes). Moreover, using survival analysis (survfit function in R package survival), we estimated the probability of returning to being susceptible to EV-A71 after natural infection.

To ensure the reproducibility, all data and codes are presented on GitHub. We have included Data Availability section and Code Availability section (Page 34, Lines 672-678).

Data Availability

The data that supporting the findings of this study are available on GitHub at <https://github.com/JYoung2022FD/Sero-epi-data>

Code Availability

The code used to generate these analyses are available on GitHub at <https://github.com/JYoung2022FD/Sero-epi-data>

2.3 Overall, the study design seems sound, but I have a few questions:

2.3.1 Were the estimates of sero-prevalence and sero-incidence corrected for imperfect sensitivity and specificity of the test?

Response: Thanks for this comment. The reviewer is right that the neutralising assays on neutralising antibodies against EV-A71 is not perfect, although it is currently used as the major antibody test and widely accepted for EV-A71¹⁻⁴. With regards to the sensitivity and specificity of the test, we searched PubMed, Web of Science, China National Knowledge Infrastructure, and Wanfang up to September 26, 2022, using the following search terms: (enterovirus 71 OR EV-A71 OR EV-71 OR EV71) AND (sero-prevalence OR seroprevalence OR sero-incidence OR sero-incidence) AND (sensitivity OR specificity OR false positive* OR false negative*). There was no estimate of sensitivity and specificity of neutralising assay for EV-A71. Accordingly, we did not adjust the assay directly.

Instead, we accounted for factors that may have an impact on the reliability of the test: 1) potential cross-reactivity between EV-A71 and other enteroviruses; 2) quality of laboratory testing procedure; and 3) the choice of antibody titre threshold for determining sero-prevalence and sero-incidence.

Firstly, previous studies found that the sera from coxsackievirus A(CVA)16 infected patients were tested negative on an EV-A71 neutralization assay⁵. Following vaccination with inactivated EV-A71, seroconversion and protection is specific for EV-A71 infection,

without similar neutralising antibodies response and cross-protection observed for other enteroviruses like CVA16, CVA6 and CVA10⁶⁻⁹. Accordingly, infections of other enteroviruses would not lead to seropositive against EV-A71 (that is, false positivity).

Secondly, to ensure accuracy of neutralising assay, a range of measures were taken in our study for quality control, which have been presented in details previously¹⁰. Here in the revised main text, we provided a summary in the Discussion section.

Page 20-21, Lines 397-405: ...As previous studies described^{37,38}, we took a series of measures for quality control of laboratory assays. For instance, each testing plate included two positive antibody control wells, two virus control wells, one serum toxicity control well for each test sample, and four cell control wells. Virus back titration was performed in each batch of test to determine the amount of attacking virus (within 32-320 TCID₅₀/50ml). EV-A71 neutralising antibody standards (strongly positive, weakly positive and negative serum) from National Institutes for Food and Drug Control were used for quality control of serum antibody titre³⁹.

Thirdly, with regards to the choice of antibody titre threshold for determining seroprevalence and sero-incidence, the protective titre of EV-A71 antibody has not been well characterized, but a phase 3 clinical trial of EV-A71 vaccines demonstrated that a titre of 16 could be considered as a possible serologic marker for protection against EV-A71-related HFMD¹¹. Moreover, a previous study showed that the choice of antibody titre threshold (i.e., 8, 16 or 32) had minimal effect on the pattern for seropositivity¹⁰. Thus, in the main analysis, seropositivity was defined as a titre of 16 or greater. New infection with EV-A71 was conservatively defined as an individual whose titres moved from below to above the infection cut-off. In the revised manuscript, sensitivity analyses have been added using a cutoff of 8 and 32.

Page 27, Lines 539-540 (Methods section): Additionally, sensitivity analyses were done with a cutoff of eight (minimum detectable antibody level in neutralisation assays) and 32.

Page 11, Lines 199-201 (Results section): The threshold of seropositivity had little impact on the estimates of seroprevalence and incidence of EV-A71 infections (Supplementary Tables 7-8).

In the revised manuscript, we have added discussions on the sensitivity and specificity of neutralising assay for EV-A71.

Pages 20-21, Lines 384-410 (Discussion section): Although neutralising assays is currently used as the major test and widely accepted for neutralising antibodies against EV-A71, some factors may have an impact on the reliability of the test and

thus on the estimates of seroprevalence and incidence of EV-A71 infections. Firstly, cross-reactivity between EV-A71 and other enteroviruses may lead to false positive of EV-A71 infections. However, previous studies found that the sera from coxsackievirus A (CVA)16 infected patients were tested negative on an EV-A71 neutralization assay ³². Following vaccination with inactivated EV-A71, seroconversion and protection is specific for EV-A71 infection, without similar neutralising antibodies response and cross-protection observed for other enteroviruses like CVA16, CVA6 and CVA10 ^{33, 34, 35, 36}.

Secondly, quality of laboratory testing procedure is critical for ensuring reliability of the test. As previous studies described ^{37, 38}, we took a series of measures for quality control of laboratory assays. For instance, each testing plate included two positive antibody control wells, two virus control wells, one serum toxicity control well for each test sample, and four cell control wells. Virus back titration was performed in each batch of test to determine the amount of attacking virus (within 32-320 TCID₅₀/50ml). EV-A71 neutralising antibody standards (strongly positive, weakly positive and negative serum) from National Institutes for Food and Drug Control were used for quality control of serum antibody titre³⁹.

Thirdly, the choice of antibody titre threshold would influence the estimates of seroprevalence and incidence of EV-A71 infections. We used a cutoff of 16 in the main analysis, and added a cutoff of eight and 32 in the sensitivity analyses. We found that our results were robust to the choice of cutoffs.

Table S7. Seroprevalence under a cutoff of 1:8, 1:16 and 1:32 for EV-A71 antibody, separately

	2013	2014	2015	2016
No. cases	2475	2135	2077	2032
Seroprevalence (cutoff 1:8)				
No. infected	1420	1337	1397	1486
Prevalence (mean, 95%CI)	57 (55, 59)	63 (61, 65)	67 (65, 69)	73 (71, 75)
Seroprevalence (cutoff 1:16)				
No. infected	1412	1326	1386	1478
Prevalence (mean, 95%CI)	57 (55, 59)	62 (60, 64)	67 (65, 69)	73 (71, 75)
Seroprevalence (cutoff 1:32)				
No. infected	1329	1247	1296	1370
Prevalence (mean, 95%CI)	54 (52, 56)	58 (56, 61)	62 (60, 64)	67 (65, 69)
Comparison amongst above three cutoffs of EV-A71 infections				
χ^2 value	8.328	9.495	13.077	20.104
P value	0. 016 [†]	0.009 [†]	0.001 [†]	< 0.001 [†]
Comparison amongst two cutoffs of EV-A71 infections (1:8 vs. 1:16)				
χ^2 value	0.040	0.100	0.109	0.061
P value [§]	0.841	0.752	0.741	0.805
Comparison amongst two cutoffs of EV-A71 infections (1:8 vs. 1:32)				
χ^2 value	6.627	7.764	10.558	15.578

P value [§]	0.010 [‡]	0.005 [‡]	0.001 [‡]	< 0.001 [‡]
--------------------	--------------------	--------------------	----------------------

Comparison amongst two cutoffs of EV-A71 infections (1:16 vs. 1:32)

χ^2 value	5.497	5.950	8,335	13.435
----------------	-------	-------	-------	--------

p value [§]	0.019	0.015 [‡]	0.004 [‡]	< 0.001 [‡]
-------	--------------------	--------------------	----------------------

Note: [§]After Bonferroni adjustment, the significance p value was adjusted from 0.05 to 0.017, which means p value <0.017 represents significant difference.

[‡] significant difference.

Table S8. Seroincidence under a cutoff of 1:8, 1:16 and 1:32 for EV-A71 antibody, separately

	2013/2014	2014/2015	2015/2016
No. cases	2136	1918	1862
Seroincidence (cutoff 1:8)			
No. seroconversion	118	103	145
Incidence (mean, 95%CI)	5.5 (4.6-6.6)	5.4 (4.4-6.5)	7.8 (6.6-9.1)
Seroincidence (cutoff 1:16)			
No. seroconversion	116	104	146
Incidence (mean, 95%CI)	5.4 (4.5-6.5)	5.4 (4.5-6.5)	7.8 (6.7-9.2)
Seroincidence (cutoff 1:32)			
No. seroconversion	124	103	143
Incidence (mean, 95%CI)	5.8 (4.9-6.9)	5.4 (4.4-6.5)	7.7 (6.5-9.0)
Comparison amongst above three cutoffs of incidence of EV-A71 infections			
χ^2 value	0.308	0.007	0.035
P value	0.857	0.997	0.983

2.3.2 Did any child in the two cohorts receive the EV-A71 vaccine? If so, could that bias the estimates of sero-prevalence?

Response: Thank you for this comment. For Children cohort (1-9 years old), no study participants were administered EV-A71 vaccines during the study period. Seven participants in Neonates cohort were vaccinated against EV-A71 after 6 months of age. The reviewer is correct that antibody titres induced by vaccination could lead to overestimating sero-prevalence in these participants. We accordingly excluded their

antibody titres after vaccination. We provide this information in the revised main text:

Page 28, Line 562-566 (Methods section): No study participants in Children cohort were administered EV-A71 vaccines during the study period. Seven study participants in Neonates cohort were administered EV-A71 vaccines after 6 months of age during the study period; thus, their antibody titres after vaccination were excluded from this analysis²³.

2.3.4 In figure 3C, please justify the choice of the arbitrary cut-off titre of 128.

Response: Thank you for this comment. We appreciate this choice appears arbitrary and we therefore now present a sensitivity analysis on that threshold. Correspondingly, we have modified the main text.

Page 29-30, Lines 585-590 (Methods section): Moreover, using survival analysis (survfit function in R package survival), we estimated the probability of returning to being susceptible to EV-A71 after natural infection. This analysis was further stratified by initial antibody titres in order to explore the difference of immunity duration. We used an initial antibody titre of 128 in the main analysis, and conducted sensitivity analyses using 256, 64 or 32 instead.

Page 14, Line 248-258 (Results section): For all naturally infected individuals at baseline, the probability of returning to being susceptible increased with age but remained low at the age of 11 years old (7.2%, 95% CI: 5.2%-9.1%). A higher probability occurred in those individuals whose initial antibody titres were <128 (5.9% [95% CI: 3.4%-8.2%] at 5 years old with a peak of 22.5% [95% CI: 16.5%-28.0%] at 11 years old) compared to those whose initial antibody titres were ≥128 (0.2% [95% CI: 0-0.5%] at 5 years old with a peak of 1.2% [95% CI: 0-2.3%] at 7 years old) (Fig. 3c). Sensitivity analyses showed that lower initial titres would lead to a large increase of the probability. For instance, in those individuals whose initial antibody titres were <64, the probability increased to 53% (95% CI: 38%, 65%) at the age of 11 years old (Supplementary Fig. 5).

Page 17-18, Lines 326-339 (Discussion section): ...However, in naturally infected individuals whose initial antibody titres were <128, the probability of returning to being susceptible increased to 22.5% at 11 years old. The probability would even increase to over 50% if initial antibody titres were <64. The antibody levels would be maintained via potential frequent re-exposures to EV-A71. Accordingly, the probability of returning to being susceptible would be underestimated. However, we were unable to evaluate the impact of potential re-exposures on susceptibility due to limited follow-up visits and long intervals between follow-up visits... To increase

vaccine uptake and ensure longer protection, we strongly recommend age-based universal vaccination instead of risk-based vaccination for EV-A71.

Figure S5. Comparison of probability of returning to be susceptible to EV-A71 in participants who were infected between groups with different initial antibody titres at baseline. Panel A: a cutoff titre of 256; Panel B: a cutoff titre of 128; Panel C: a cutoff titre of 64; Panel D: a cutoff titre of 32.

2.3.4 To calculate sero-incidence, the authors write that: “New infection with EV-A71 was conservatively defined as an individual whose titres moved from below to above the infection cut-off.” (lines 174–176). Please also report this information in the methods, and explain how the confidence intervals for infection rates (Figure 2b) were derived. It would also be useful to assess how sensitive the estimates of infection rates were to the assumed cut-off.

Response: Thanks for pointing this out. We have added the information as follows:

Page 27, Lines 536-540 (Methods section): ..., in the main analysis, seropositivity was defined as a titre of 16 or greater. New infection with EV-A71 was conservatively defined as an individual whose titres moved from below to above the infection cutoff. Additionally, sensitivity analyses were done with a cutoff of eight (minimum detectable antibody level in neutralisation assays) and 32.

Page 29, Lines 571-572 (Methods section): We used a binomial distribution to derive the 95% confidence intervals (using R package binom).

Page 11, Lines 199-201 (Results section): The threshold of seropositivity had little impact on the estimates of seroprevalence and incidence of EV-A71 infections (Supplementary Tables 7-8).

Page 21, Lines 407-410 (Discussion section): Thirdly, the choice of antibody titre threshold would influence the estimates of seroprevalence and incidence of EV-A71 infections. We used a cutoff of 16 in the main analysis, and added a cutoff of eight and 32 in the sensitivity analyses. We found that our results were robust to the choice of cutoffs.

2.4 From figure 3, the authors conclude that antibody decay and return to susceptibility are slow (except for children whose initial antibody titres were low). However, a general limitation of sero-prevalence studies (even with detailed longitudinal follow-up as this one) is that not all events of **re-exposure** are observed. Hence, it could be that the “natural” decay rate of antibodies is quite high, but that apparently high levels of antibodies (as observed in figure 3B) are maintained via frequent re-exposures to EV-A71. This limitation may be especially important in the context of vaccination, as vaccines can reduce overall circulation in the population, and therefore also the frequency of re-exposure.

Response: Thank you for this comment. The reviewer raises a good point. We have revised the discussion as follows:

Page 17-18, Lines 320-339 (Discussion section): Our findings reveal that naturally infected individuals have a low probability of returning to a susceptible status. From this perspective, the findings support China CDC's recommendation for the target population of EV-A71 vaccination. That is, susceptible individuals (namely, those without a protective neutralising antibody titre against EV-A71-related diseases) aged 6 months-5 years⁹. Hereafter it is referred to a risk-based vaccination recommendation. However, in naturally infected individuals whose initial antibody titres were <128, the probability of returning to being susceptible increased to 22.5% at 11 years old. The probability would even increase to over 50% if initial antibody titres were <64. The antibody levels would be maintained via potential frequent re-exposures to EV-A71. Accordingly, the probability of returning to being susceptible would be underestimated. However, we were unable to evaluate the impact of potential re-exposures on susceptibility due to limited follow-up visits and long intervals between follow-up visits. In addition, the risk-based vaccination recommendation would have a negative impact on the implementation of vaccination programs (i.e., facing the challenge of identifying susceptible individuals) and thus on vaccine coverage²⁶. To increase vaccine uptake and ensure longer protection, we strongly recommend age-based universal vaccination instead of risk-based vaccination for EV-A71.

2.5 From a more general, public health perspective, I wonder if the study results are sufficient to support the authors' claim, as stated in the conclusions: "Our findings support completing vaccination before 1 year of age, and we recommend age-based universal vaccination for birth cohorts and a one-time catch-up vaccination for children 2-5 years of age." (lines 331-334).

Response: Thanks for pointing this out. Per reviewer 1's comment 1.2, we have added more detailed analyses for the first year of life, particularly before 6 months of age, to address the ideal age for incorporation of EV-A71 vaccine into the National Immunization Program. We found that as maternal EV-A71 antibody declined rapidly, over half of children were susceptible to EV-A71 at 1 month of age, while the proportion increased to three quarters at 2 months of age and then peaked after 5 months of age. Correspondingly, incidence of EV-A71 infections and EV-A71-related HFMD showed an increasing trend before 6 months of age, and incidence of EV-A71 infections peaked at 5-6 months of age. Our previous study demonstrated that the time to loss of protective immunity of maternal antibody was less than two months¹⁰. Moreover, children younger than 6 months have higher risk of severe outcomes given infections relative to older children¹².

We strongly recommend further investigating the optimal vaccination timing of approved EV-A71 vaccines in mainland China as early as 1-2 months of age if introduced into the National Immunisation Programme, through further assessment the dosing, safety, and effectiveness¹⁰. In addition, if EV-A71 vaccines are introduced into the National Immunization Programme for children younger than 6 months of age on the basis of a birth cohort, a one-time catch-up vaccination for children 6 months-5 years of age is highly recommended, accounting for their high risk of EV-A71 infection and EV-A71–related HFMD due to susceptibility and increasing frequent contact with other children.

Correspondingly, we revised the Results and Discussion. Details were presented in our response to comment 1.2. Moreover, we revised our conclusions:

Page 22, lines 432-435: Our findings support completing vaccination before 6 months of age, and we recommend age-based universal vaccination for birth cohorts and a one-time catch-up vaccination for children 6 months-5 years old.

2.5.1 First, there is an obvious extrapolation problem (acknowledged by the authors), as the study was only conducted in rural areas of Southern China and the epidemiology of EV-A71 may differ in other regions.

Response: Thanks for this comment. The reviewer is right that the absolute number of seroprevalence and incidence of EV-A71 infections in southern China would be different from that of northern China and other countries, as well as that of other years outside the study period, accounting for the variations of intensity and seasonal characteristics of EV-A71 activity between regions and years. Accordingly, it is not possible to generalize our estimates of absolute values. However, we hypothesise that the age patterns of EV-A71 infections identified by serology may be similar between southern and northern China given similar age patterns of incidence rates for HFMD notifications¹³. The age pattern is key for informing the appropriate age of vaccination. In this case, our recommendations on age-based universal vaccination for birth cohorts (completing vaccination before 6 months of age) and a one-time catch-up vaccination for children 6 months-5 years old would be reliable in other regions, which merits further studies in the future. We have revised the corresponding discussion in the main text.

Page 19, lines 356-374: In southern China, two peaks are observed per year (in spring and autumn), whereas only one peak is observed in summer in northern China². EV-A71 has been circulating in the Asian-Pacific region since the 1990s. EV-A71 activity has remained at a low level in Europe and the USA for decades, but the number of EV-A71–related outbreaks has increased in the past several years^{27, 28, 29, 30}. Moreover, the predominant enterovirus serotype associated with HFMD varies between years^{3, 31}. Given these different histories of EV-A71 exposure, it is not

possible to generalize our estimates on seroprevalence and incidence of EV-A71 infections in southern China to northern China and other countries, as well as to other years outside the study period. However, the age patterns of EV-A71 infections identified by serology may be similar between southern and northern China, given similar age patterns of incidence rates for HFMD notifications³ as well as similar age patterns of EV-A71 seroprevalence shown in a systematic review²¹. The age pattern is key for informing the appropriate age of vaccination. In this case, our recommendations on age-based universal vaccination for birth cohorts (completing vaccination before 6 months of age) and a one-time catch-up vaccination for children 6 months-5 years old would be appropriate in other regions, although this merits further studies in the future.

2.5.2 Second, ultimately the decision to introduce a new vaccine should rely on other indicators, such as hospitalization or mortality rates.

Response: Thanks for pointing this out. In addition to seroprevalence as well as incidence of EV-A71 infections and EV-A71-related HFMD, our recommendations of vaccination also account for the case-severity risk and case-fatality risk. We revised the Discussion section as below:

Pages 16, lines 300-304: ...Moreover, children younger than 6 months have higher risk of severe outcomes (e.g., case-fatality risk: about 0.17% in children younger than 6 months vs 0.11% in those aged 6–11 months) given infections relative to older children ². Altogether, administering the first doses at age 6 months might be too late to protect infants younger than that.

Page 16-17, lines 312-318: ...if EV-A71 vaccines are introduced into the National Immunization Programme for children younger than 6 months of age on the basis of a birth cohort, a one-time catch-up vaccination for children 6 months-5 years of age is highly recommended, accounting for their high risk of EV-A71 infection and EV-A71-related HFMD due to susceptibility and increasing frequent contact with other children, as well as high case-severity risk and case-fatality risk ².

2.5.3 Figure 2C presents incidence rates of EV-A71-related severe HFMD, but the definition of “severe” in this case remained unclear, after re-reading the corresponding description (lines 416–424)—please clarify.

Response: Thanks for your comments. Figure 2C presents the total incidence rates of EV-A71 HFMD of all severity, not only severe cases.

2.5.4 Finally, one should be aware of the indirect effects of targeted vaccination, especially in the context of disease caused by polymorphic pathogens like EV. For example, one could imagine that the different EV types compete with each other, such that targeted vaccination against EV-71 could cause a release of other types. Not being in the field of EV epidemiology, I can't assess if this issue is important (or, indeed, relevant at all), but I think it should be, at least, discussed.

Response: Thanks for your comments. The reviewer raises a good point. We have revised Discussions as suggested.

Page 21-22, lines 412-427: Moreover, several studies have found that the predominant enterovirus serotype associated with HFMD was CVA16 or other enteroviruses, such as CVA6, other than EV-A71, between 2013 and 2016 in Anhua and across China^{3, 40}. Between 2008 and 2018, EV-A71 was replaced by other enterovirus serotypes that were the predominant serotypes for HFMD in 2013, 2015, 2017 and 2018^{3, 31}. In 2018, for the first time, other enterovirus serotypes replaced EV-A71 as the predominant serotype for severe HFMD in China³¹. Moreover, a mathematical model study found that a potential high coverage of EV-A71 vaccination is likely to lead to transient and minor serotype replacement by CVA16⁴¹. Additional efforts are therefore required to characterize the full burden of HFMD in children. The residual sera of our cohorts are of great value to address the seroepidemiological characteristics of other enterovirus serotypes, and relevant cross-immunity. In addition, longer-term surveillance of seroepidemiological characteristics of EV-A71 and other enteroviruses in larger geographic regions is recommended to help adjust and refine the vaccination strategy if necessary.

Minor comments

2.6 The abstract would need to incorporate further information about the study design, period, and location. Please also report uncertainty intervals for the sero-prevalence estimates.

Response: Thanks for pointing this out. We have added the information as follows:

Page 4, lines 53-58: ...we studied the seroprevalence and annual infection incidence of EV-A71, and quantified the dynamics of neutralising antibodies, with prospective population-based cohorts of children established in Anhua County, China. We randomly enrolled 4188 children aged 1-9 years, and 1066 pairs of neonates and mothers since Autumn 2013, and followed them up for three years.

Page 4, lines 61-65: ...with 56.3% (95% CI: 52.6%-59.9%) of neonates became susceptible at 1 month of age. Between 6 months and 2 years old, over 80% of study participants were susceptible to EV-A71, while 63.8% (95% CI: 60.6%-67.0%) and 34.4% (95% CI: 31.1%-37.8%) remained susceptible at 3 and 5 years old, respectively.

2.7 p4, lines 55–57: it seems these two sentences should not be separated by a full stop.

Response: Thanks for your suggestion. We have revised as suggested.

2.8 p13, l 247: I don't think the word "naivety" actually exists in English.

Response: "Naivety" is defined as "lack of experience, wisdom, or judgement" in the Oxford Languages dictionary.

Reviewer #3 (Remarks to the Author):

3.1 In the abstract it would be helpful to state the data used. It isn't clear whether the incidence estimates are coming from the seroprevalence or different data. Also the study design is a strength of the paper, so good to highlight here!

Response: Thanks for your comments. We have revised as suggested as follows:

Page 4, Lines 53-59: ...we studied the seroprevalence and annual infection incidence of EV-A71, and quantified the dynamics of neutralising antibodies, with prospective population-based cohorts of children established in Anhua County, China. We randomly enrolled 4188 children aged 1-9 years, and 1066 pairs of neonates and mothers since Autumn 2013, and followed them up for three years. Neutralising antibodies against EV-A71 were measured in 59% of children aged 1-9 years and all neonates.

3.2 Line 59-60: I think stating for what proportion of the population this is the case would be helpful.

Response: Thanks for your suggestion. We have revised as suggested as follows:

Page 4, Lines 68-69: Although geometric mean titre after natural infection declined with age, the seroprevalence remained above 90% at 12 years old.

3.3 Line 59: a word missing or incorrect tense I think. With the recommendation of

vaccination at one year old, I think it would be good to highlight in the abstract the seropositivity at age 1.

Response: Thanks for pointing it out. Per reviewer 1's comment 1.2, we have added more detailed analyses for the first year of life, particularly before 6 months of age, to address the ideal age for incorporation of EV-A71 vaccine into the National Immunization Program. Accordingly, in the revised manuscript, we recommended EV-A71 vaccination for children younger than 6 months of age on the basis of a birth cohort, and a one-time catch-up vaccination for children 6 months-5 years of age. Per your comments, we have revised as follows:

Pages 4-5, Lines 60-73: As the maternal antibody titres declining rapidly, the susceptible individuals accumulated, with 56.3% (95% CI: 52.6%-59.9%) of neonates became susceptible at 1 month of age. Between 6 months and 2 years old, over 80% of study participants were susceptible to EV-A71, while 63.8% (95% CI: 60.6%-67.0%) and 34.4% (95% CI: 31.1%-37.8%) remained susceptible at 3 and 5 years old, respectively. The highest incidence of EV-A71 infections was observed in children 5-6 months, while the highest EV-A71-related HFMD incidence was observed in children 1-2 years old. Although geometric mean titre after natural infection declined with age, the seroprevalence remained above 90% at 12 years old. Lower initial titres would lead to a large increase of the probability. The results support completing EV-A71 vaccination before 6 months of age. Here we recommend age-based universal vaccination for birth cohorts and a one-time catch-up vaccination for children 6 months-5 years old.

3.4 Line 120-121, I think the phrasing about "rural areas" is maybe a little general without further discussion of how similar you think this transmission would be across different rural areas. Suggest either rephrasing here or adding more information on the (lack of) variation.

Response: Thanks for pointing it out. We have removed "rural areas" and added more discussions with regards to the extrapolation of our findings.

Pages 19, lines 356-374: In southern China, two peaks are observed per year (in spring and autumn), whereas only one peak is observed in summer in northern China². EV-A71 has been circulating in the Asian-Pacific region since the 1990s. EV-A71 activity has remained at a low level in Europe and the USA for decades, but the number of EV-A71-related outbreaks has increased in the past several years^{27, 28, 29, 30}. Moreover, the predominant enterovirus serotype associated with HFMD varies between years^{3, 31}. Given these different histories of EV-A71 exposure, it is not

possible to generalize our estimates on seroprevalence and incidence of EV-A71 infections in southern China to northern China and other countries, as well as to other years outside the study period. However, the age patterns of EV-A71 infections identified by serology may be similar between southern and northern China, given similar age patterns of incidence rates for HFMD notifications³ as well as similar age patterns of EV-A71 seroprevalence shown in a systematic review²¹. The age pattern is key for informing the appropriate age of vaccination. In this case, our recommendations on age-based universal vaccination for birth cohorts (completing vaccination before 6 months of age) and a one-time catch-up vaccination for children 6 months-5 years old would be appropriate in other regions, although this merits further studies in the future.

3.5 Line 217: Is this V shape hypothesized to start with maternal immunity, if so, please mention here. I understand this has been looked at by a previous paper by the authors, but I think also worth discussing here. Particularly with the conclusions around timing of vaccination- it will be important to vaccinate after the decline in maternal immunity too I would think.

Response: Thanks for your suggestion. As stated previously, we have added more detailed analyses for the first year of life, particularly before 6 months of age on the basis of Neonate cohort, to address the ideal age for incorporation of EV-A71 vaccine into the National Immunization Program. We have revised as suggested:

Page 13, Lines 237-239: Starting with maternal immunity, a V-shape was observed for neutralising antibody titres of EV-A71 before 5 years old in all study participants, and titres tended to level off afterwards (Fig. 3a).

Moreover, we have modified the Discussion section to discuss the ideal age of initial vaccination and catch-up vaccination if introducing the EV-A71 vaccine into the National Immunization Program. Details were presented in our response to comment 1.2.

3.6 Line 219, be useful to say natural infection, instead of what? Or how natural infection is defined.

Response: Thanks for your comments. We have revised as suggested as follows:

Pages 13-14, Lines 243-245: Among all seropositive individuals, after natural infection other than vaccination, GMT declined from 737 (95% CI: 520-1044) to 80 (95% CI: 62-103) with age (Fig. 3b).

3.7 Figure 2, Panel B: This is the estimated incidence from the serology? Please clarify.

Response: Thanks for pointing it out. We have revised as suggested as follows:

Page 42, Lines 882-884: Panel B: **b** Age-specific incidence of EV-A71 infections identified by serology for those study participants with paired sera before and after HFMD epidemics²³.

General comments

3.8 Please comment on any/the lack of possible cross reactivity.

Response: Thanks for your comments. We have added discussions as suggested.

Page 20, Lines 387-394: ...cross-reactivity between EV-A71 and other enteroviruses may lead to false positive of EV-A71 infections. However, previous studies found that the sera from coxsackievirus A (CVA)16 infected patients were tested negative on an EV-A71 neutralization assay³². Following vaccination with inactivated EV-A71, seroconversion and protection is specific for EV-A71 infection, without similar neutralising antibodies response and cross-protection observed for other enteroviruses like CVA16, CVA6 and CVA10^{33, 34, 35, 36}.

3.9 Be helpful to have more comparison to the age distribution of cases and any different inferences that would be made on incidence of infection from the age distribution of cases.

Response: Thanks for pointing this out. Per the reviewer 1's comment 1.2, we have added more detailed analyses for the first year of life, by breaking down into 0-2, 3-4, 5-6, and 7-11 months of age groups (Figure 2C). We have revised the Results section as below:

Pages 12-13, Lines 221-230: High incidence rates of EV-A71 infection were observed before 5 years of age, which showed an increasing trend before 6 months of age and then declined, but remained very high in children aged 3-5 years old (Fig. 2b)... Similar pattern was observed for incidence rates of EV-A71-related HFMD, which increased over age and peaked at 2 years old, and then declined but remained high at 3-5 years old (Fig. 2c).

3.10 Figure 2 (Panel A): The older ages susceptible proportions don't seem to be

captured so well by the model- what is happening here?

Response: Thanks for pointing this out. Several factors may lead to this phenomenon: 1) sample sizes were smaller at older age groups, so the variations were greater than those in younger groups. For example, there were only 67 observations at 12 years of age. 2) We used age-specific (in months) proportions of susceptible individuals to fit the model, but grouped the observed proportions by age of years for clarity. The differences of age scales could contribute to the divergence between observed points and the fitted curve. 3) The model we selected had minimum AIC but was constituted with a spline of degree 2, which was less smooth than that of degree 3. Therefore, we chose the model with a spline of degree 3 and the lowest AIC, and redrew the plot as follows.

Figure 2 (Panel a). Age-specific proportion of susceptible populations for all study participants in Children and Neonate cohort (points represent observed mean proportion, whereas vertical lines represent corresponding 95% CI; blue curve represents fitted mean proportion, whereas blue shadow represents corresponding 95% CI).

We have added details in the Supplementary Notes as follows:

We applied generalized linear mixed models (PROC glimmix in SAS) using B-splines to fit the dynamics of the proportion of susceptible individuals. The model selection including B-spline's degree and knots, and model parameters were based on Akaike Information Criterion (AIC). The original model with a spline of degree two had minimum AIC. However, the observed proportions of susceptible individuals in older age groups were not be captured well by the model. Moreover, the degree of splines is commonly set as three since this degree can depict smooth and natural curves ²⁸.

Therefore, we built a spline of degree three to characterize the gradual change of the susceptible proportion. The residual deviations between observed proportions of susceptible individuals and fitted curve in older age groups were caused by below factors: 1) sample sizes were smaller at older age groups, so the variations were greater than those in younger groups. For example, there were only 67 observations at 12 years of age. 2) We used age-specific (in months) proportions of susceptible individuals to fit the model, but grouped the observed proportions by age of years for clarity.

References

1. Kuo FL, *et al.* Seroprevalence of EV-A71 neutralizing antibodies following the 2011 epidemic in HCMC, Vietnam. *PLoS neglected tropical diseases* **14**, e0008124 (2020).
2. Yang B, *et al.* Seroprevalence of Enterovirus 71 Antibody Among Children in China: A Systematic Review and Meta-analysis. *The Pediatric infectious disease journal* **34**, 1399-1406 (2015).
3. Luo ST, *et al.* Enterovirus 71 maternal antibodies in infants, Taiwan. *Emerging infectious diseases* **15**, 581-584 (2009).
4. Ooi EE, Phoon MC, Ishak B, Chan SH. Seroepidemiology of human enterovirus 71, Singapore. *Emerging infectious diseases* **8**, 995-997 (2002).
5. Shih SR, *et al.* Expression of Caspid Protein VP1 for Use as Antigen for the Diagnosis of Enterovirus 71 Infection. *Journal of medical virology* **61**, 228-234 (2000).
6. Jiang H, *et al.* The epidemiological characteristics of enterovirus infection before and after the use of enterovirus 71 inactivated vaccine in Kunming, China. *Emerg Microbes Infec* **10**, 619-628 (2021).
7. Lim H, *et al.* The immunogenicity and protection effect of an inactivated coxsackievirus A6, A10, and A16 vaccine against hand, foot, and mouth disease. *Vaccine* **36**, 3445-3452 (2018).
8. Chong P, Liu CC, Chow YH, Chou AH, Klein M. Review of Enterovirus 71 Vaccines. *Clinical Infectious Diseases* **60**, 797-803 (2015).
9. Mao Q, *et al.* The compatibility of inactivated-Enterovirus 71 vaccination with Coxsackievirus A16 and Poliovirus immunizations in humans and animals. *Hum Vaccin Immunother* **11**, 2723-2733 (2015).
10. Wei X, *et al.* The transfer and decay of maternal antibodies against enterovirus A71, and dynamics of antibodies due to later natural infections in Chinese infants: a longitudinal, paired mother-neonate cohort study. *Lancet Infectious Diseases* **21**, 418-426 (2021).
11. Zhu F, *et al.* Efficacy, safety, and immunology of an inactivated alum-adjuvant enterovirus 71 vaccine in children in China: a multicentre, randomised, double-blind, placebo-controlled, phase 3 trial. *Lancet* **381**, 2024-2032 (2013).
12. Xing W, *et al.* Hand, foot, and mouth disease in China, 2008-12: an

epidemiological study. *Lancet Infect Dis* **14**, 308-318 (2014).

13. Yang B, *et al.* Epidemiology of hand, foot and mouth disease in China, 2008 to 2015 prior to the introduction of EV-A71 vaccine. *Euro Surveill* **22**, pii=16-00824 (2017).

REVIEWERS' COMMENTS

Reviewer #1 (Remarks to the Author):

The authors have responded sufficiently to all reviewer comments and the manuscript is greatly improved.

Reviewer #2 (Remarks to the Author):

Reviewer’s report on: “Seroepidemiology of children with enterovirus A71 infections in southern China: longitudinal, population-based cohort studies, 2013-2018”

I thank the authors for their detailed answers to my comments. My technical comments have been addressed; though not the authors’ fault, the absence of sensitivity and specificity estimates in the literature is surprising and may weaken this study (for example low specificity would result in over-estimates of EV-A71 seroprevalence).

I am less pleased with the authors’ reply to my last major comment, about whether the evidence presented justifies their recommendation of universal vaccination in China. The corresponding sentences in the revised abstract have not been changed (“The results support completing EV-A71 vaccination before 6 months of age. Here we recommend age-based universal vaccination for birth cohorts and a one-time catch-up vaccination for children 6 months-5 years old.” p 4–5), and in their reply the authors merely acknowledged that my concerns were valid. In fact, the authors’ reply has made these concerns even more severe.

First, regarding the geographical extrapolation issue, the cited study about the epidemiology of Hand, Foot, and Mouth Disease (HFMD) in 2018 in China shows large spatial variability in the distribution of enterovirus serotypes, with very low prevalence of EV-A71 in some northern regions (such as Qinhai, figure S1 in Ref. [1]). The authors comment that the age distribution of HFMD is comparable in southern and northern regions of China; but this seems beside the point, as the pre-vaccine distribution of enterovirus serotypes is also expected to predict the impact of EV-A71 vaccination.

Second, regarding the risk of competitive release of other serotypes following vaccination against EV-A71, the same paper indicates that serotypes other than EV-A71 now represent the vast majority of lab-confirmed HFMD in China (Figure 1C). In their reply, the authors cite a modeling study that found a minor risk of replacement by serotype CV-A16 [2]. However, this study focused only on serotypes EV-A71 and CV-A16, and did find evidence of cross-protection between the two. Hence, similar cross-protection may exist for other serotypes (which are collectively more prevalent than CV-A16 [1]), such that the risk of competitive release cannot be ruled based on this study alone.

All in all, the authors’ claim that their data justify universal vaccination seems unfounded, and further studies would be needed to support it. As I wrote in my first review, the study is well-conducted and provides useful estimates of EV-A71 sero-prevalence, but is in itself insufficient to have implications for vaccine recommendation.

References

- [1] Liu F, Ren M, Chen S, Nie T, Cui J, Ran L, et al. Pathogen Spectrum of Hand, Foot, and Mouth Disease Based on Laboratory Surveillance - China, 2018. *China CDC Wkly.* 2020 Mar;2(11):167-71.
- [2] Takahashi S, Liao Q, Van Boeckel TP, Xing W, Sun J, Hsiao VY, et al. Hand, Foot, and Mouth Disease in China: Modeling Epidemic Dynamics of Enterovirus Serotypes and Implications for Vaccination. *PLoS Med.* 2016 Feb;13(2):e1001958.

Reviewer’s report on: “Seroepidemiology of children with enterovirus A71 infections in southern China: longitudinal, population-based cohort studies, 2013-2018”

I thank the authors for their detailed answers to my comments. My technical comments have been addressed; though not the authors’ fault, the absence of sensitivity and specificity estimates in the literature is surprising and may weaken this study (for example low specificity would result in over-estimates of EV-A71 seroprevalence).

I am less pleased with the authors’ reply to my last major comment, about whether the evidence presented justifies their recommendation of universal vaccination in China. The corresponding sentences in the revised abstract have not been changed (“The results support completing EV-A71 vaccination before 6 months of age. Here we recommend age-based universal vaccination for birth cohorts and a one-time catch-up vaccination for children 6 months-5 years old.” p 4–5), and in their reply the authors merely acknowledged that my concerns were valid. In fact, the authors’ reply has made these concerns even more severe.

First, regarding the geographical extrapolation issue, the cited study about the epidemiology of Hand, Foot, and Mouth Disease (HFMD) in 2018 in China shows large spatial variability in the distribution of enterovirus serotypes, with very low prevalence of EV-A71 in some northern regions (such as Qinhai, figure S1 in Ref. [1]). The authors comment that the age distribution of HFMD is comparable in southern and northern regions of China; but this seems beside the point, as the pre-vaccine distribution of enterovirus serotypes is also expected to predict the impact of EV-A71 vaccination.

Second, regarding the risk of competitive release of other serotypes following vaccination against EV-A71, the same paper indicates that serotypes other than EV-A71 now represent the vast majority of lab-confirmed HFMD in China (Figure 1C). In their reply, the authors cite a modeling study that found a minor risk of replacement by serotype CV-A16 [2]. However, this study focused only on serotypes EV-A71 and CV-A16, and did find evidence of cross- protection between the two. Hence, similar cross-protection may exist for other serotypes (which are collectively more prevalent than CV-A16 [1]), such that the risk of competitive release cannot be ruled based on this study alone.

All in all, the authors’ claim that their data justify universal vaccination seems unfounded, and further studies would be needed to support it. As I wrote in my first review, the study is well-conducted and provides useful estimates of EV-A71 seroprevalence, but is in itself insufficient to have implications for vaccine recommendation.

References

- [1] Liu F, Ren M, Chen S, Nie T, Cui J, Ran L, et al. Pathogen Spectrum of Hand, Foot, and Mouth Disease Based on Laboratory Surveillance - China, 2018. *China CDC Wkly.* 2020 Mar;2(11):167-71.

- [2] Takahashi S, Liao Q, Van Boeckel TP, Xing W, Sun J, Hsiao VY, et al. Hand, Foot, and Mouth Disease in China: Modeling Epidemic Dynamics of Enterovirus Serotypes and Implications for Vaccination. *PLoS Med.* 2016 Feb;13(2):e1001958.

Response: We apologize that we did not fully understand the reviewer's previous comment. We would like to thank the reviewer for further assessing our manuscript and the constructive comments. With his/her further explanations, we agree that our findings are not sufficient for universal vaccination recommendation. Accordingly, we revised the Discussion and Conclusion as follows:

Page 18-19, lines 354-373: In southern China, two HFMD peaks are observed per year (in spring and autumn), whereas only one peak is observed in summer in northern China². EV-A71 has been circulating in the Asian-Pacific region since the 1990s. EV-A71 activity has remained at a low level in Europe and the USA for decades, but the number of EV-A71-related outbreaks has increased in the past several years^{28, 29, 30, 31}. Additionally, the predominant enterovirus serotype associated with HFMD varies between years^{3, 32}. EV-A71 was replaced by CVA16 or other enteroviruses serotypes that were the predominant serotypes for HFMD in several years (like 2013 and 2018) in China^{3, 32, 33}. Given these different histories of EV-A71 exposure, it is not possible to generalize our estimates on seroprevalence and incidence of EV-A71 infections in southern China to northern China and other countries, as well as to other years outside the study period. Furthermore, a mathematical model study found that a potential high coverage of EV-A71 vaccination is likely to lead to transient and minor serotype replacement by CVA16³⁴. Longer-term surveillance of enteroviruses in larger geographic regions is therefore required to depict the seroepidemiological characteristics of EV-A71 and other enteroviruses, which could help selecting vaccine antigen, adjust and refine the vaccination strategy if necessary. Moreover, the development of multivalent vaccines could certainly contribute to prevention and control of HFMD³⁵.

Page 20, lines 405-413: This finding suggests in southern China, completing

vaccination before 6 months of age would be beneficial, and we recommend age-based vaccination for birth cohorts. The findings that large proportions of children 6 months-5 years of age remained susceptible and their high incidence of EV-A71-related HFMD indicate a potentially one-time catch-up vaccination is of value in this age group. More regionally representative longitudinal studies, such as this study, that follow individuals with serial serology over multiple seasons for a longer period of time are needed to further validate these findings.

Due to word limit (150 words), we re-organized the Abstract, and revised the conclusions in the Abstract as follows:

Page 4, lines 60-63: Our findings suggest EV-A71 vaccination before 6 months for birth cohorts in southern China, potentially with a one-time catch-up vaccination for children 6 months-5 years old. More regionally representative longitudinal seroepidemiological studies are needed to further validate these findings.